# Backdoor Federated Learning by Poisoning Backdoor-Critical Layers

Haomin Zhuang[1], Mingxian Yu[1],[*] Hao Wang[2], Yang Hua[3], Jian Li[4], Xu Yuan[5]

[1]South China University of Technology [2]Louisiana State University
[3]Queen's University Belfast, UK [4]Stony Brook University [5]University of Delaware
{z365460860,mxyu1112}@gmail.com, haowang@lsu.edu,
Y.Hua@qub.ac.uk,jian.li.3@stonybrook.edu,xyuan@udel.edu,

## Abstract

Federated learning (FL) has been widely deployed to enable machine learning training on sensitive data across distributed devices. However, the decentralized learning paradigm and heterogeneity of FL further extend the attack surface for backdoor attacks. Existing FL attack and defense methodologies typically focus on the whole model. None of them recognizes the existence of *backdoor-critical (BC) layers*—a small subset of layers that dominate the model vulnerabilities. Attacking the BC layers achieves equivalent effects as attacking the whole model but at a far smaller chance of being detected by state-of-the-art (SOTA) defenses. This paper proposes a general in-situ approach that identifies and verifies BC layers from the perspective of attackers. Based on the identified BC layers, we carefully craft a new backdoor attack methodology that adaptively seeks a fundamental balance between attacking effects and stealthiness under various defense strategies. Extensive experiments show that our BC layer-aware backdoor attacks can successfully backdoor FL under seven SOTA defenses with only 10% malicious clients and outperform latest backdoor attack methods.

## 1 Introduction

Federated learning (FL) (McMahan et al., 2017) enables machine learning across large-scale distributed clients without violating data privacy. However, such decentralized learning paradigm and heterogeneity in data distribution and client systems extensively enlarge FL's attack surface. Increasing numbers of attack methods have been developed to either slow down the convergence of FL training (*i.e.*, *untargeted attacks* (Fang et al., 2020; Baruch et al., 2019; Shejwalkar & Houmansadr, 2021; El El Mhamdi et al., 2018)) or enforce the model to intentionally misclassify specific categories of data (*i.e.*, *targeted attacks* (Xie et al., 2019; Bagdasaryan et al., 2020; Bhagoji et al., 2019; Wang et al., 2020; Li et al., 2023)).

As a subset of targeted attacks, *backdoor attacks* (Xie et al., 2019; Bagdasaryan et al., 2020; Wang et al., 2020; Gu et al., 2017; Li et al., 2023) are one of the stealthiest attacks for FL, which train models on data with special *triggers* embedded, such as pixels, textures, and even patterns in the frequency domain (Feng et al., 2022). Models compromised by backdoor attacks typically have high accuracy on general data samples (*i.e.*, main task) except that samples with triggers embedded activate the "backdoor" inside the model (*i.e.*, backdoor task), leading to misclassification targeted to specific labels (*e.g.*, recognizing a stop sign as a speed limit sign).

Several defense methods have been proposed to detect backdoor attacks and mitigate their impacts, which can be classified into three types based on their key techniques: *distance-based*, *inversion-based*, and *sign-based* defense. Distance-based defenses, such as FLTrust (Cao et al., 2021) and FoolsGold (Fung et al., 2020), calculate the cosine similarity distance and euclidean distance between the local models to detect potential malicious clients. Inversion-based defenses, such as Zhang et al. (2022a), utilize trigger inversion and backdoor unlearning to mitigate backdoors in global models. Sign-based defenses, such as RLR (Ozdayi et al., 2021), detect the sign change directions of each parameter in the local model updates uploaded by clients and adjust the learning rate of each parameter to mitigate backdoor attacks. Therefore, existing backdoor attacks can hardly work around the detection reinforced by the aforementioned multi-dimension defenses.

---

[*]This work was performed when Haomin Zhuang and Mingxian Yu were remote intern students advised by Dr. Hao Wang at the LSU IntelliSys Lab.

We have observed a new dimension ignored by existing studies—the effectiveness of backdoor attacks is only related to a small subset of model layers—*backdoor-critical (BC) layers*. To demonstrate the existence of BC layers, we first train a benign five-layer CNN model on a clean dataset until it has converged. Then, we train a copy of the benign model on poisoned data (with triggers embedded) and obtain a malicious model. We substitute each layer in the benign model for the same layer in the malicious model and measure the backdoor attack success rate, which denotes the accuracy of recognizing samples with trigger embedded as the targeted label. Fig. 1(a) shows that the absence of layers in the malicious model does not degrade the BSR except for the `fc1.weight` layer. Fig. 1(b) shows the reversed layer substitution that only the `fc1.weight` layer from the malicious model enables successful backdoor tasks. Therefore, we argue that a small set of layers, such as `fc1.weight`, are *backdoor-critical*—the absence of even one BC layer leads to a low Backdoor Success Rate. BC layers as a small subset of models can be observed in large models like ResNet18 and VGG19 (refer to Fig. A-22 ). Intuitively, deeper layers are more BC because shallower layers learn simple, low-level features such as edges and textures, and deeper layers combine these features to learn more complex, high-level concepts such as objects and their parts (Zeiler & Fergus, 2014; Bau et al., 2017; Simonyan et al., 2013).

This paper proposes a Layer Substitution Analysis, a general in-situ approach that identifies BC layers using forward and backward layer substitutions. We further craft two new backdoor attack methods: layer-wise poisoning attack and layer-wise flipping attack. These two backdoor attack methods leverage the identified BC layers to bypass state-of-the-art (SOTA) distance-based, inversion-based, and sign-based defense methods by carefully attacking the BC layers with minimal model poisoning and a small number of clients (*i.e.*, 10% of the participating clients). Our contributions include:

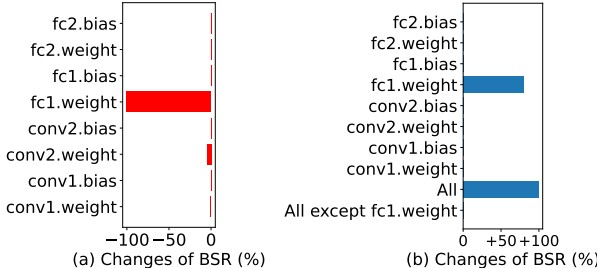

Figure 1: (a) The changes in backdoor success rate (BSR) of the malicious model with a layer substituted from the benign model. (b) The changes of BSR of the benign model with layer(s) substituted from the malicious model ("All except `fc1.weight`" indicates replacing all layers except the `fc1.weight` with layers from the malicious model).

- We propose Layer Substitution Analysis, a novel method that recognizes backdoor-critical layers, which naturally fits into FL attackers' context.
- We design two effective layer-wise backdoor attack methods, that successfully inject backdoor to BC layers and bypass SOTA defense methods without decreasing the main task accuracy.
- Our evaluation on a wide range of models and datasets shows that the proposed layer-wise backdoor attack methods outperform existing backdoor attacks, such as DBA (Xie et al., 2019), on both main task accuracy and backdoor success rate under SOTA defense methods.

## 2 PRELIMINARIES

### 2.1 FEDERATED LEARNING (FL)

FL leverages a large set of distributed clients, denoted as $\mathcal{N} = \{1, \ldots, N\}$, to iteratively learn a global model $\boldsymbol{w}$ without leaking any clients' private data to the central coordinator server (McMahan et al., 2017). Formally, the objective is to solve the following optimization problem:

$$\min_{\boldsymbol{w}} F(\boldsymbol{w}) := \sum_{i \in \mathcal{N}} p^{(i)} f_i(\boldsymbol{w}^{(i)}),$$

where $f_i(\boldsymbol{w}^{(i)}) = \frac{1}{|D^{(i)}|} \sum_{(x,y) \in D^{(i)}} \ell(x, y; \boldsymbol{w}^{(i)})$ is the local objective function of $i$-th client with its local dataset $D^{(i)}$, and $p^{(i)} = |D^{(i)}|/\sum_{i \in \mathcal{N}} |D^{(i)}|$ is the relative data sample size. FL training process solves this optimization problem by aggregating local models from distributed clients to update the global model iteratively.

### 2.2 THREAT MODEL OF BACKDOOR ATTACKS

**Attacker's goal:** As the existing studies on FL backdoor attacks (Gu et al., 2017; Bagdasaryan et al., 2020; Xie et al., 2019; Ozdayi et al., 2021; Wang et al., 2020), an attacker's goal is to enforce models

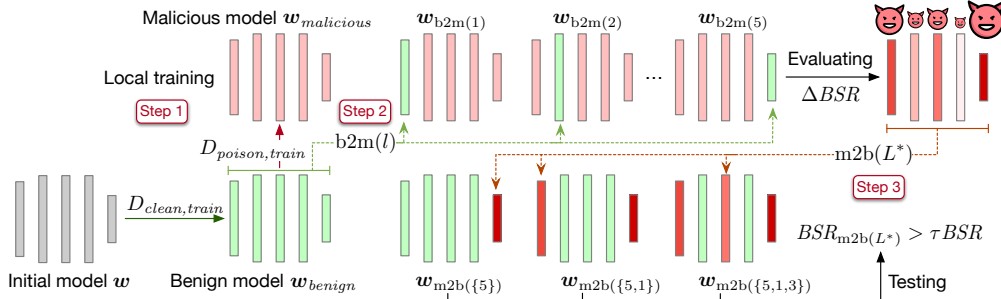

Figure 2: Identifying BC layers with Layer Substitution Analysis. $b2m(l)$ indicates inserting the $l$,-*th* layer in the benign model to the malicious model, $m2b(L^*)$ indicates inserting the malicious model's layers within the subset $L^*$ to the benign model, and BSR indicates Backdoor Success Rate.

to classify data samples with triggers embedded to specific incorrect labels (*i.e.*, the backdoor task), while keeping a high accuracy for samples without triggers embedded (*i.e.*, the main task).

**Attacker's capabilities:** We assume that an attacker compromises a subset $\mathcal{M} = \{1, \ldots, M\}$ of malicious clients. However, the proportion of malicious clients is assumed to be less than 50%, *i.e.*, $|\mathcal{M}|/|\mathcal{N}| < 50\%$. Otherwise, existing FL defense methods can hardly withstand such backdoor attacks. Following existing studies (Fung et al., 2020; Fang et al., 2020; Bagdasaryan et al., 2020; Baruch et al., 2019; Yin et al., 2018; Ozdayi et al., 2021; Nguyen et al., 2021), malicious clients controlled by the attacker can communicate with each other to synchronize attacking strategies. The attacker also has access to a snapshot of the global model in each round and can directly manipulate model weights and datasets on each malicious client (Fang et al., 2020; Li et al., 2023).

## 3 IDENTIFYING BC LAYERS

In the FL setting, there is a global model in each turn, where clients can train on their local data with a few epochs, which the models in clients are similar to each other so that the new global models are from averaging all clients' models (Konečnỳ et al., 2016). So in this setting, we have an opportunity to explore the difference between the malicious models that are trained on some poison dataset and benign that are trained on the clean dataset.

We argue that if the $l$-th layer (or a subset $L^*$ of layers) is critical to the backdoor task, substituting the layer(s) in $\boldsymbol{w}_{benign}$ with the same layer(s) in $\boldsymbol{w}_{malicious}$ will cause a decline in the accuracy of backdoor task.

### 3.1 OVERVIEW

Fig. 2 presents the Layer Substitution Analysis to identify BC layers for each malicious client $i$ controlled by the attacker, where $i \in \mathcal{M}$:

**Step 1:** The initial model $\boldsymbol{w}$ is trained on the clean dataset $D_{clean,\ train}$ to obtain a benign model $\boldsymbol{w}_{benign}$. Then, the benign model $\boldsymbol{w}_{benign}$ is further trained on the poison dataset $D_{poison,\ train}$ to converge for learning backdoor task to obtain the malicious model $\boldsymbol{w}_{malicious}$.

**Step 2:** Forward layer substitution—substituting the individual $l$-th layer of the malicious model $\boldsymbol{w}_{malicious}$ with the $l$-th layer of the benign model $\boldsymbol{w}_{benign}$ iteratively, where $l \in L$. Then we evaluate the backdoor success rate (BSR) of the updated malicious model $\boldsymbol{w}_{b2m(l)}$ and compare it with the BSR of the original malicious model $\boldsymbol{w}_{malicious}$. We sort the layers by the changes in BSR.

**Step 3:** Performing backward layer substitution following the order of layers sorted in Step 2. We incrementally copy the layer from the malicious model to the benign model until the BSR of the updated model reaches a threshold. Then, the indexes of the copied layers denote the set of BC layers $L^*$.

### 3.2 LAYER SUBSTITUTION ANALYSIS

**Step 1: Local Training** In FL setting, malicious clients identify BC layers on their local datasets. A local dataset $D^{(i)}$ in i-*th* malicious client is split into training sets $D^{(i)}_{clean,train}$ and $D^{(i)}_{poison,train}$ as well as validation sets $D^{(i)}_{clean,val}$ and $D^{(i)}_{poison,val}$. Upon a malicious client $i$ receives the global model

$\boldsymbol{w}$, it trains the benign model $\boldsymbol{w}_{benign}$ from the global model on the clean dataset $D^{(i)}_{clean,train}$ until it converges. Then, the attacker trains the $\boldsymbol{w}_{benign}$ on the poisoned dataset $D^{(i)}_{poison,train}$ to converge to obtain a malicious model $\boldsymbol{w}_{malicious}$.

**Step 2: Forward Layer Substitution** We argue that if a layer is BC, replacing it in the malicious model with a "benign" layer from the benign model will decrease the malicious model's backdoor task accuracy (BSR).

**Benign layer $\to$ malicious model**: We first examine the change of BSR when the malicious model is replaced with a layer from the benign model at each client $i$. Specifically, we use $b2m(l)$ to denote the process that replaces a malicious model's $l$-th layer with a benign model's $l$-th layer, where both models have the same structure, including $|L|$ layers ($L$ denotes the set of layers).

As Fig. 2 shows, executing $b2m(l)$ generates an updated malicious model $\boldsymbol{w}_{b2m(l)}$ per layer replacement. We then evaluate the BSR of the updated malicious models $\boldsymbol{w}_{b2m(l)}$, $l \in L$ with the poisoned dataset $D_{poison,\,val}$. By iterating through all layers $l \in L$, each malicious client $i$ can sort the layers according to the change of BSR, defined as:

$$\Delta BSR_{b2m(l)} := BSR_{malicious} - BSR_{b2m(l)},$$

where $BSR_{malicious}$ denotes the BSR of the poisoned model $\boldsymbol{w}_{malicious}$, and $BSR_{b2m(l)}$ denotes the BSR of the updated model $\boldsymbol{w}_{b2m(l)}$, which has the $l$-th layer replaced. With the layers sorted by the $\Delta BSR_{b2m(l)}$ from the highest to the lowest, we further perform backward layer substitution to confirm the identification of BC layers.

**Step 3: Backward Layer Substitution** We argue that if a layer is BC, replacing it in the benign model with a "malicious" layer from the malicious model will increase the BSR of the benign model.

**Malicious layers $\to$ benign model**: The backward layer substitution process is defined as $m2b(L^*)$. Unlike $b2m(l)$ that only replaces an individual layer, $m2b(L^*)$ replaces a subset $L^*$ of layers. We iteratively add a layer into $L^*$ following the descending order of $\Delta BSR_{b2m(l)}$ and evaluate the BSR of the updated model with the poisoned dataset $D_{poison,\,val}$. Fig. 2 shows $m2b(L^*)$ iteratively copies the subset $L^*$ of layers from the malicious model $\boldsymbol{w}_{malicious}$ to the benign model $\boldsymbol{w}_{benign}$ until $BSR_{m2b(L^*)}$ reaches a pre-defined threshold $\tau BSR_{malicious}$, where $\tau \in (0,1]$ and $BSR_{m2b(L^*)}$ denotes the BSR of the updated model $\boldsymbol{w}_{m2b(L^*)}$. Specifically, we compare $BSR_{m2b(L^*)}$ with the threshold $\tau BSR_{malicious}$ as follows:

If $BSR_{m2b(l)} < \tau BSR_{malicious}$, we should add another layer following the descending order of $\Delta BSR_{b2m(l)}$ to $L^*$ and execute $m2b(L^*)$ to update the model $\boldsymbol{w}_{m2b(L^*)}$. Then, we re-evaluate the BSR of the updated model on the poisoned dataset $D_{poison,\,val}$ and compare it with the threshold again.

If $BSR_{m2b(l)} \geq \tau BSR_{malicious}$, the new model $\boldsymbol{w}_{m2b(L^*)}$ has achieved a similar BSR as the malicious model $BSR_{malicious}$. We stop adding more layers to $L^*$.

Then, we argue that the layers in the subset $L^*$ are BC since these layers satisfy both conditions: 1) removing them from the malicious model decreases its BSR. 2) copying them to the benign model increases its BSR to a similar rate as the malicious model. It should be noted that backward layer substitution can identify individual BC layers and BC combinations of layers (*i.e.*, the backdoor task is jointly learned by a combination of layers).

## 4 POISONING BC LAYERS

The identified BC layers provide a new perspective to craft more precise and stealthy backdoor attacks on FL. This section presents two attack methods with awareness of backdoor-critical layers: **layer-wise poisoning (LP) attack** that attacks both *distance-based* and *inversion-based* defense methods and **layer-wise flipping (LF) attack** that attacks *sign-based* defense methods.

### 4.1 LAYER-WISE POISONING (LP) ATTACK

With the subset $L^*$ of identified BC layers, we design LP attack that selects BC layers from $L^*$ and precisely poisons the layers with minimal modification, which can bypass existing distance-based defense methods (Cao et al., 2021; Nguyen et al., 2021; Blanchard et al., 2017).

In $t$-th round, malicious clients selected by the FL server perform forward layer substitution and backward layer substitution to find out the set $L^*_t$ of BC layers. After receiving the global model $\boldsymbol{w}_t$

(we denote $\boldsymbol{w}$ as $\boldsymbol{w}_t$ for simplicity), malicious client $i$ trains two local models $\boldsymbol{w}_{malicious}^{(i)}$ and $\boldsymbol{w}_{benign}^{(i)}$ with its local dataset $D_{poison}^{(i)}$ and $D_{clean}^{(i)}$, respectively.

We propose a vector $\boldsymbol{v}=[\boldsymbol{v}_1, \boldsymbol{v}_2, ..., \boldsymbol{v}_l]$ to denote the selection of the subset from the model $\boldsymbol{w}_{benign}^{(i)}$ or $\boldsymbol{w}_{malicious}^{(i)}$. If $\boldsymbol{v}_j = 1$, the $j$-th layer of the benign model $\boldsymbol{w}_{benign}^{(i)}$ will be substituted with the corresponding layer in the malicious model $\boldsymbol{w}_{malicious}^{(i)}$. We next introduce $\boldsymbol{u}_{malicious}^{(i)}=[\boldsymbol{u}_{malicious,1}^{(i)}, \boldsymbol{u}_{malicious,2}^{(i)}, .., \boldsymbol{u}_{malicious,l}^{(i)}]$ to denote the model $\boldsymbol{w}_{malicious}^{(i)}$ in layer space, where $\boldsymbol{u}_{malicious,j}^{(i)}$ is the $j$-th layer in the model. $\boldsymbol{u}_{benign}^{(i)}$ denotes the model $\boldsymbol{w}_{benign}^{(i)}$ layer-wisely in the same way. The goal of the attacker in round $t$ is formulated as an optimization problem:

$$\max_{\boldsymbol{v}} \quad \frac{1}{\left|D^{(i)}\right|} \sum_{(x,y)\in D^{(i)}} P[G(x') = y'; \boldsymbol{w}_{t+1}], \tag{1}$$

$$\text{s.t. } \boldsymbol{w}_{t+1} = \mathcal{A}(\widetilde{\boldsymbol{w}}^{(1)}, \ldots, \widetilde{\boldsymbol{w}}^{(M)}, \boldsymbol{w}^{(M+1)}, \cdots, \boldsymbol{w}^{(N)}), \tag{2}$$

$$\widetilde{\boldsymbol{w}}^{(i)} = \boldsymbol{v} \circ \boldsymbol{u}_{malicious}^{(i)} + (1 - \boldsymbol{v}) \circ \boldsymbol{u}_{benign}^{(i)}, \tag{3}$$

where $\circ$ denotes the element-wise multiplication, $\boldsymbol{w}_{t+1}$ denotes global model weight in round $t+1$, $\mathcal{A}$ denotes the aggregation function in the server, $x'$ denotes a image embedded with trigger, $y'$ denotes the targeted label, and $G(x)$ denotes the predicted label of global model with input $x$. Aggregation functions $\mathcal{A}$ can utilize clustering algorithm K-means or HDBSCAN, making it infeasible to calculate gradients.

To address this optimization challenge, we propose a straightforward approach. In order to conform to the constraints, the attacker must perform adaptive attacks by adjusting the number of layers targeted during each round of the attack. Following previous work (Fang et al., 2020), attacker can estimate the models in benign clients using the local benign models on the malicious clients. These locally-available benign models can then be utilized to simulate the selection process on the server, through the initialization of the crafted model $\widetilde{\boldsymbol{w}}^{(i)}$ with a subset $L^*$ obtained through Layer Substitution Analysis. When the crafted model $\widetilde{\boldsymbol{w}}^{(i)}$ is rejected during the simulation, attacker decrease the size of the subset $L^*$ by removing layers in the order in which they are added to the set in backward layer substitution process. To further minimize the distance, attacker uses the model averaged from those local benign models $\boldsymbol{u}_{average} = \frac{1}{M}\sum_{k=0}^{M}\boldsymbol{u}_{benign}^{(k)}$ to make $\widetilde{\boldsymbol{w}}^{(i)}$ closer to the center of benign models. Then we introduce a hyperparameter $\lambda \geq 0$ to control the stealthiness of the attack:

$$\widetilde{\boldsymbol{w}}^{(i)} = \lambda\boldsymbol{v} \circ \boldsymbol{u}_{malicious}^{(i)} + ReLU(1 - \lambda) \cdot \boldsymbol{v} \circ \boldsymbol{u}_{average} + (1 - \boldsymbol{v}) \circ \boldsymbol{u}_{average}, \tag{4}$$

where $ReLU(x) = x$ if $x > 0$ and $ReLU(x) = 0$, otherwise, and it is similar to Scaling attack when $\lambda > 1$.

For defenses that lack filter-out strategies like FedAvg, attacker can assume that the server is implementing strict distance-based strategies, such as FLAME and MultiKrum, to solve the optimization problem within this framework. The identification is not necessary for each round, attacker can identify BC layers in any frequency, like every 5 rounds. The analysis of the trade-off between frequency and backdoor success rate refers to §5.4.

## 4.2 LAYER-WISE FLIPPING (LF) ATTACK

When LP attack fails to bypass sign-based defense methods, the backdoor-related parameters probably reside in the non-consensus sign regions and are neutralized by learning rates with reversed signs. To work around such sign-based defense methods, we propose a Layer-wise Flipping attack that keeps the efficacy of BC layers by proactively flipping the parameters signs of the layers in $L^*$ on each client $i$ before the defense methods apply a reversed learning rate to the layers, defined as:

$$\boldsymbol{w}_{LFA}^{(i)} := -(\boldsymbol{w}_{m2b(L^*)}^{(i)} - \boldsymbol{w}) + \boldsymbol{w}.$$

Eventually, the parameters of BC layers are flipped by the FL server again, which restores the sign of the parameters and activates the backdoor injected into the model. With the knowledge of BC layers, Layer-wise Flipping attack avoids unnecessarily poisoning the other layers, which improves the main task accuracy and disguises the malicious updates from being detected by the defense methods.

Table 1: Detection accuracy of FLAME and MultiKrum on CIFAR-10 dataset. MAR indicates malicious clients acceptance rate (%), and BAR indicates benign clients acceptance rate (%).

| Models (Dataset) | Attack | MultiKrum non-IID | | FLAME non-IID | | MultiKrum IID | | FLAME IID | |
|---|---|---|---|---|---|---|---|---|---|
| | | MAR | BAR | MAR | BAR | MAR | BAR | MAR | BAR |
| VGG19 (CIFAR-10) | Baseline | 10.1 | 43.28 | 16.58 | 73.54 | 0.5 | 44.39 | 0.0 | 69.11 |
| | LP Attack | **91.0** | **34.33** | **93.0** | **59.39** | **99.5** | **33.34** | **100** | **55.67** |
| | DBA | 0.5 | 44.39 | 12.25 | 74.1 | 0.5 | 44.39 | 0.08 | 68.61 |
| ResNet18 (CIFAR-10) | Baseline | 3.0 | 44.11 | 5.58 | 74.86 | 0.0 | 44.44 | 0.17 | 72.95 |
| | LP Attack | **93.01** | **34.11** | **93.0** | **59.39** | **94.35** | **33.97** | **99.0** | **58.83** |
| | DBA | 0.5 | 44.39 | 3.5 | 75.06 | 0.0 | 44.44 | 0.17 | 72.55 |
| CNN (Fashion-MNIST) | Baseline | 0.0 | 44.44 | 0.25 | 66.81 | 0.0 | 44.44 | 0.0 | 66.78 |
| | LP Attack | **78.11** | **35.77** | **100.0** | **55.67** | **68.13** | **36.87** | **99.0** | **55.67** |
| | DBA | 0.0 | 44.44 | 0.5 | 67.11 | 0.0 | 44.44 | 0.0 | 66.69 |

## 5 EVALUATION

We implement Layer Substitution Analysis and the two attack methods by PyTorch (Paszke et al., 2019). We conduct all experiments using a NVIDIA RTX A5000 GPU. By default, we use 100 clients in FL training, while 10% of them are malicious. In each round 10% clients are selected to train models locally. The non-IID dataset are sampled as $q = 0.5$ following Cao et al. (2021). Please refer to §11 for the details of experiments settings.

### 5.1 METRICS

*Acc* denotes the main task accuracy of the converged global model on the validation dataset. ***Backdoor success rate (BSR)*** is the proportion that the global model successfully mis-classifies images with triggers embedded to the targeted labels. ***Benign-client acceptance rate (BAR)*** and ***malicious-client acceptance rate (MAR)*** indicate the accuracy of defense strategies detecting malicious clients. BAR denotes the proportion of benign models accepted by defense aggregation strategies among all benign models uploaded by benign clients. MAR denotes the proportion of malicious clients accepted by defense aggregation strategies.

### 5.2 THE ATTACKS' STEALTHINESS

Table 1 shows that MultiKrum and FLAME successfully prevent most malicious updates by the baseline attack and DBA since their MARs are approximating to zero. Besides, the large gap between MARs and BARs of the baseline attack and DBA indicates that MultiKrum and FLAME easily distinguish malicious and benign updates when selecting updates for aggregation.

However, the high MAR achieved by LP attack indicates it successfully has its malicious updates accepted by the FL server running MultiKrum and FLAME. LP attack bypasses the detection of MultiKrum and FLAME on all settings. Besides, the gap between LP Attack's MAR and BAR indicates that malicious updates are more likely to be accepted as benign ones by the server.

To further demonstrate the stealthiness of LP attack, we plot the Krum distance in BadNets attack, Scaling attack, and LP attack in ResNet18 trained on IID CIFAR-10. The sum of square distance is denoted as Krum distance. A large Krum distance means the model update is far from other local model updates and less likely to be accepted by the server. Malicious model updates from LP attack are close to benign model updates, which causes the failure of MultiKrum detection.

Fig. A-9 plots participant clients' Krum distance in each 5 rounds, which shows that it is hard for the defense strategy to distinguish malicious updates attacked by LP attack from benign ones. Scaling attack presents larger Krum distances than BadNets attack, so we do not consider Scaling attack as a normal baseline attack in our experiments.

### 5.3 THE ATTACKS' EFFECTIVENESS

Table 2 shows that LP attack achieves the highest Acc (*i.e.*, main task accuracy) and the highest BSR under most settings. Fig. 3 illustrate that the convergence rate of the backdoor task using the LP attack is generally faster than the baseline attack across various settings. We can observe the similar results in IID settings in Table A-6 and Fig. A-11 in Appendix.

Notably, for large models such as VGG19 and ResNet18 on CIFAR-10, LP attack is successful in embedding the backdoor, while the baseline attack and DBA fail in FLAME (IID and non-IID), MultiKrum (IID and non-IID), FLDetector (IID), and FLARE(IID). Even in the scenario of Multi-

Table 2: Main task accuracy and BSR on Non-IID datasets. We mark the BSR below 10% (corresponding to ten classes of the datasets) as red, indicating a failed attack, and mark the highest BSR as **bold** within the same setting. The Baseline is BadNets (Gu et al., 2017). The results are the average of five repeated experiments. For LP attack (LF attack), $a \pm b$, where $a$ is the mean value, and $b$ is the standard deviation. Acc: main task accuracy (%), Avg: average, BSR unit: %.

| Model (Dataset) | | VGG19 (CIFAR-10) | | | ResNet18 (CIFAR-10) | | | CNN (Fashion-MNIST) | | |
|---|---|---|---|---|---|---|---|---|---|---|
| Attack | | Baseline | LP Attack (LF Attack) | DBA | Baseline | LP Attack (LF Attack) | DBA | Baseline | LP Attack (LF Attack) | DBA |
| FedAvg (non-IID) | Best BSR | 84.88 | **92.8**±0.99 | 41.15 | 85.19 | **94.19**±0.99 | 21.19 | 99.97 | 87.69±4.3 | **99.97** |
| | Avg BSR | 74.69 | **83.55**±0.43 | 25.88 | 70.53 | **89.12**±1.4 | 10.94 | 99.9 | 78.84±9.16 | **99.9** |
| | Acc | 78.89 | **79.95**±0.46 | 78.97 | 77.58 | 77.89±0.43 | **77.99** | 88.28 | **88.42**±0.23 | 87.95 |
| FLTrust (non-IID) | Best BSR | **92.91** | 76.56±34.38 | 42.14 | **92.43** | 82.05±25.34 | 37.16 | 74.17 | 89.44±3.44 | **100.0** |
| | Avg BSR | 67.3 | 65.44±31.56 | 15.88 | **75.84** | 71.52±29.17 | 15.11 | 68.97 | 77.05±4.67 | **100.0** |
| | Acc | 75.1 | 74.03±4.06 | **75.11** | 75.72 | 69.9±5.74 | **77.51** | **89.51** | 89.48±0.1 | 89.31 |
| FLAME (non-IID) | Best BSR | 47.03 | **88.68**±4.98 | 38.25 | 23.04 | **95.41**±0.93 | 9.77 | 0.18 | **84.33**±3.12 | 0.58 |
| | Avg BSR | 7.78 | **60.72**±2.44 | 7.33 | 7.22 | **90.15**±3.51 | 3.88 | 0.1 | **74.91**±2.66 | 0.4 |
| | Acc | 62.91 | 56.92±1.12 | **63.3** | **76.04** | 71.48±0.36 | 75.27 | 87.78 | 87.05±0.21 | **87.89** |
| RLR (non-IID) | Best BSR | 79.37 | **92.17**±1.81 (2.79±0.81) | 43.79 | 81.61 | **93.16**±0.85 (1.37±0.02) | 13.85 | 20.27 | 0.0 ± 0.0 (**70.52**±3.13) | 38.25 |
| | Avg BSR | 74.01 | **89.24**±2.09 (0.6±0.09) | 33.69 | 60.83 | **82.14**±7.46 (0.7±0.1) | 7.8 | 15.09 | 0.0 ± 0.0 (**66.12**±2.94) | 7.33 |
| | Acc | 67.33 | **72.1**±0.58 (63.2±3.94) | 64.3 | 75.07 | 73.44±0.95 (**76.48**±0.32) | 75.04 | 85.56 | 86.09 ± 0.13 (**86.45**±0.41) | 63.3 |
| MultiKrum (non-IID) | Best BSR | 22.93 | **95.87**±0.51 | 29.44 | 12.72 | **95.94**±0.97 | 10.63 | 1.09 | **89.95**±2.74 | 0.28 |
| | Avg BSR | 7.84 | **75.93**±2.49 | 8.44 | 3.95 | **90.12**±1.38 | 5.61 | 0.39 | **74.94**±6.97 | 0.1 |
| | Acc | 58.93 | **69.28**±3.29 | 64.81 | **74.49** | 72.26±1.34 | 73.02 | 87.31 | **87.58**±0.21 | 87.58 |
| FLDetector (non-IID) | Best BSR | 95.49 | 87.28±0.69 | 16.28 | 5.23 | **90.31**±2.04 | 5.89 | 74.64 | 99.45±0.13 | **99.93** |
| | Avg BSR | **95.42** | 86.71±0.54 | 16.14 | 5.21 | **86.56**±1.32 | 5.87 | 66.11 | 96.32±0.41 | **99.9** |
| | Acc | 55.25 | **57.95**±1.37 | 56.67 | **64.39** | 63.89±0.91 | 65.25 | 79.16 | 75.96±0.81 | **79.78** |
| FLARE (non-IID) | Best BSR | **96.67** | 93.47±4.32 | 25.48 | 17.16 | **79.94**±4.06 | 26.96 | 2.02 | 82.64±4.16 | **100** |
| | Avg BSR | **94.45** | 70.23±5.83 | 8.18 | 6.24 | **53.72**±7.73 | 6.62 | 1.54 | 78.18±2.41 | **100** |
| | Acc | 70.25 | **77.28**±1.46 | 69.95 | **71.39** | 70.84±1.63 | 64.22 | 88.29 | 88.07±0.46 | 88.01 |

····· Acc under baseline attack    ——— Acc under LP attack    ·-○-· BSR under baseline attack    —○— BSR under LP attack

(a) FLAME  (b) FLTrust  (c) MultiKrum  (d) FLDetector

Figure 3: VGG19 trained with different robust aggregation rules on non-IID data. Acc indicates the main task accuracy, and BSR indicates Backdoor Success Rate.

Krum (non-IID), where LP attack shows fluctuations in BSR as illustrated in Fig. 3(c), the average BSR is still up to 76.85%, indicating that the attack is successful in most rounds.

The sign-based defense method RLR fails to reverse the signs of parameters in large models, thus LF attack fails to embed the backdoor by reversing the signs.

Fig. 3 and Fig. A-11 in the Appendix present the training progress of VGG19 on non-IID and IID data, respectively. The figures show that LP attack outperforms the baseline attack in most cases in terms of the main task accuracy and BSR. For small model CNN, FLAME (IID and non-IID), RLR (IID and non-IID), MultiKrum (IID and non-IID), FLDetector (IID), and FLARE (IID) effectively defend against both baseline attacks and DBA attacks. However, the LP attack is successful in bypassing all distance-based defense strategies in both IID and non-IID settings. The LF Attack is also effective in circumventing sign-based defense method RLR, resulting in an increase of 35% (IID) and 50% (non-IID) in BSR compared to the baseline attack.

## 5.4 SENSITIVITY ANALYSIS

**BC Layer Identification Threshold $\tau$:** We conduct a sensitivity analysis of $\tau$, which is the BC layer identification threshold, by training ResNet18 with IID datasets under FLAME and MultiKrum protection. The average BSR in Fig. 4 shows that LP attack is consistently effective under different $\tau$ values. Larger $\tau$ indicates more layers identified as BC layers and included in $L^*$, leading to a higher risk of being detected due to more layers being attacked. The adaptive layer control can

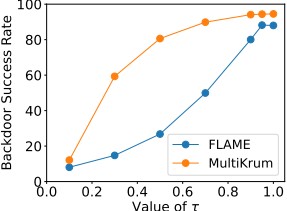

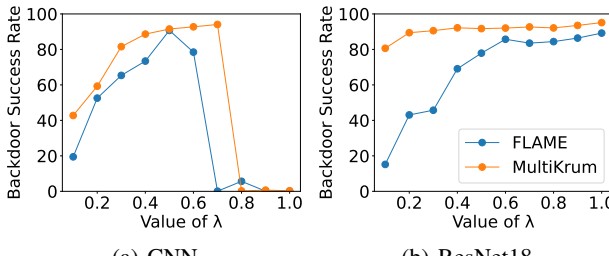

Figure 4: BSR v.s. different $\tau$ values used by LP attack.

(a) CNN          (b) ResNet18

Figure 5: Different values of $\lambda$ in CNN trained on Fashion-MNIST and ResNet18 trained on CIFAR-10.

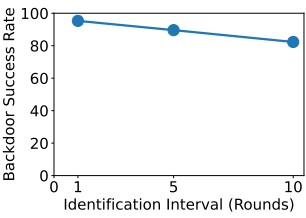

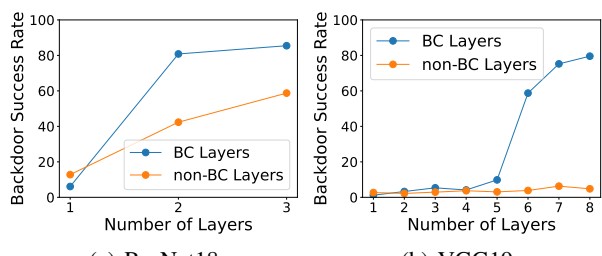

Figure 6: Impacts of different BC layers identification intervals.

(a) ResNet18          (b) VGG19

Figure 7: Attacking a fixed number of BC layers or non-BC layers under FLAME training ResNet18 on IID CIFAR-10 dataset.

decrease the number of attacking layers properly to bypass detection, which makes sure LP attack keep effective when $\tau$ is high.

**Stealthiness Knob $\lambda$:** A sensitivity analysis of the parameter $\lambda$, which governs the stealthiness of attacks, is performed by utilizing CNN and ResNet18 with IID datasets Fashion-MNIST and CIFAR-10, respectively, under the FLAME and MultiKrum defence. Fig. 5 demonstrates that the LP attack method attains the highest level of success when $\lambda = 0.5$ for the CNN model and $\lambda = 1$ for the ResNet18 model. In CNN experiments, LP attack is detected when $\lambda > 0.7$ in MultiKrum and $\lambda > 0.6$ in FLAME.

**The Impact of Identification Interval:** In our experiments, attacker identifies BC layers in each round, which is computationally expensive. Although the set of BC layers is varying with the process of FL, the sets in each round are similar. So attacker can reuse the BC layers in previous rounds or even in the first round only. We conduct experiments on ResNet18 trained on IID CIFAR-10 dataset under FedAvg. The results in Fig. 6 show that higher frequency can achieve higher BSR. The BSR is 37.9% if always reusing first-round identification. In practice, the attacker can select the frequency of identification based on their device capabilities.

## 5.5 ABLATION STUDY

**Importance of BC layers:** To show how BC layers work in Layer-wise Poisoning (LP) attack, we design a control group—Random Layer-wise Poisoning attack—that malicious clients randomly choose the same number of non-BC layers in LP attack to craft model $\widetilde{\boldsymbol{w}}^{(i)}$.

Table 3: Ablation study on BC layers in FLAME.

| Model | Attack | BSR (%) | MAR (%) |
|---|---|---|---|
| VGG19 | Baseline | 2.58 | 16.58 |
| | LP Attack | 83.86 | 100 |
| | Random LP Attack | 3.36 | 98.5 |
| ResNet18 | Baseline | 3.33 | 0.17 |
| | LP Attack | 89.9 | 100 |
| | Random LP Attack | 46.48 | 98.5 |

We evaluate the LP and Random LP attacks under FLAME by training the VGG19 and ResNet18 models on CIFAR-10 in IID setting.

Fig. 7 shows that attacking the same number of BC layers always achieves better performance in BSR, especially in VGG19. The results presented in Table 3 explain that the primary reason for the failure of baseline attacks is the low acceptance rate of malicious models, with only 16.58% and 0.17% accepted models for ResNet18 and VGG19, respectively. In contrast, the primary limitation of the Random Layer-wise Poisoning attack is its incorrect choice of model parameters, despite its high malicious acceptance rate of 98.5%. The failure of Random LP attack highlights the importance of BC layers for achieving successful backdoor tasks.

**Impact of the model averaging and adaptive change of layers:** The average model $u_{average}$ in Equation equation 4 and adaptive layer control are two mechanisms introduced in §4.1 to improve the ability of malicious models to mimic benign models, thus enabling them to evade detection by defenses. In order to demonstrate the efficacy of the LP attack, experiments are conducted both with and without these mechanisms. The results presented in Table 4 indicate that these mechanisms significantly contribute to deceiving defense strategies by increasing both the selection rate of malicious models and BSR. Notably, both mechanisms have a great impact on MAR. For instance, in VGG19 trained on non-IID CIFAR-10, model averaging increases MAR from 51% to 76%, while adaptive control rises MAR from 51% to 66%. These mechanisms are capable of working collaboratively to further improve the MAR to 93%.

Table 4: Training on CIFAR-10 dataset with (✓) and without (×) Average Model and Adaptive Layer Control.

| Distribution | Model | Model Averaging | Adaptive Control | MAR (%) | BSR (%) |
|---|---|---|---|---|---|
| non-IID | ResNet18 | ✓ | ✓ | 93.01 | 90.74 |
| non-IID | ResNet18 | ✓ | × | 76.0 | 87.43 |
| non-IID | ResNet18 | × | ✓ | 66.48 | 93.36 |
| non-IID | ResNet18 | × | × | 51.8 | 87.63 |

**Further Evaluation in Appendix:** In §12, we illustrate the superiority of the LP attack over SRA (Qi et al., 2022) by significantly increasing the BSR from approximately 4% to 96%. Additionally, we outperform Constrain Loss Attack (Li et al., 2023) by achieving a substantial margin of 82% BSR in MultiKrum and demonstrate how LP attack corrupts Flip (Zhang et al., 2022a) and achieve about 60% BSR in average. In §13, we show LP attack attains an approximate 80% BSR under low accuracy conditions in BC layers identification scenarios. In §14, we exhibit the LP attack's ability to evade adaptive layer-wise defense mechanisms, achieving no less than a 52% BSR. In §15, we show that the LP attack can successfully inject backdoor attacks even when only 0.02 of the clients are malicious. In §16, we provide evidence that our LP attack performs effectively in datasets characterized by a high degree of non-IID with a parameter $q = 0.8$.

## 6 RELATED WORK

**Subnet Attack**: Several studies, such as Bai et al. (2020); Rakin et al. (2019; 2020; 2021), inject backdoor by flipping limited bits in the computer memory. Qi et al. (2022) selects a path from the input layer to the output layer to craft a subnet that activates for backdoor tasks only. However, those attacks can be detected by FL defenses as they pay limited attention to their distance budget.

**Memorization in Training Data**: Stephenson et al. (2021); Baldock et al. (2021) believe deep layers are responsible for the memorization of training datasets. However, Maini et al. (2023) finds that the learning of noisy data in training datasets not related to the specific layers utilizing Layer Rewinding to detect the decrease in the accuracy of the noisy training dataset, which is similar to our forward layer substitution. The difference between our conclusions and Maini et al. (2023) may lie in the different tasks, where we train models to link a trigger with a specific label but Maini et al. (2023) train model to a set of "hard to learn" data, which requires more modification on parameters.

**More Related Works.** There are a variety of previous studies related to our work. We provide more detailed discussion on related works in §18.

## 7 LIMITATION AND CONCLUSION

**Limitation** Single-shot attack (Bagdasaryan et al., 2020) has the capability to inject a backdoor into the global model through a malicious client within a single round by scaling the parameters of malicious models. While our LP attack can narrow the distance gap by targeting BC layers, we acknowledge that it may not effectively support a large scaling parameter, such as $\lambda = 100$ in DBA, when confronted with stringent distance-based defenses. However, there are several possible methods to improve the LP attack to support larger scaling parameters, e.g., searching BC neurons or designing triggers related to fewer parameters.

**Conclusion** This paper proposes Layer Substitution Analysis, an algorithm that verifies and identifies the existence of backdoor-critical layers. We further design two layer-wise backdoor attack methods, LP Attack and LF Attack that utilize the knowledge of backdoor-critical layers to craft effective and stealthy backdoor attacks with minimal model poisoning. We evaluate the relationship between backdoor tasks and layers under an extensive range of settings and show our attacks can successfully bypass SOTA defense methods and inject backdoor into models with a small number of compromised clients.

## 8 ACKNOWLEDGEMENTS

The work of H. Wang was supported in part by the National Science Foundation (NSF) grants 2153502, 2315612, 2327480, and the AWS Cloud Credit for Research program. The work of J. Li was supported in part by the NSF grants 2148309 and 2315614, and the U.S. Army Research Office (ARO) grant W911NF-23-1-0072. The work of X. Yuan was supported in part by the NSF grants 2019511, 2348452, and 2315613. Any opinions, findings, and conclusions or recommendations expressed in this material are those of the authors and do not necessarily reflect the views of the funding agencies.

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

# Appendix
## Backdoor Federated Learning by Poisoning Backdoor-critical Layers

## 9  COMPLEXITY ANALYSIS

To calculate the complexity of Layer Substitution Analysis, we begin by assuming the time complexity of a single sample processed by the neural network during the forward and backward steps to be $t_{forward}$ and $t_{backward}$. Based on this assumption, the training complexity of a benign model in Step 1 of Fig. 2 can be expressed as $O(e \cdot n \cdot (t_{forward} + t_{backward}))$, where $n$ denotes the size of the training dataset and $e$ denotes the epoch needed for model convergence. Assuming that the model can converge, epoch $e$ will be a constant, and the training complexity of the benign model can be rewritten to $O(n \cdot (t_{forward} + t_{backward}))$, which is equivalent to the training complexity of a malicious model in Step 2. The insertion of a single layer from the benign model to the malicious model in Steps 3 and 4 and the corresponding testing on the test dataset have the same complexity of $O(n \cdot l \cdot t_{forward})$, where $l$ represents the number of layers in the model.

Therefore, the total complexity of the proposed Layer Substitution Analysis algorithm is $O(2n \cdot t_{backward} + 2n \cdot (l + 1) \cdot t_{forward})$, where $n$ is the sample size in the dataset. While this complexity analysis provides a baseline estimate of the algorithm's computational requirements, actual time complexity may vary due to various factors such as the neural network architecture, hyperparameters in the training process, and hardware used for training.

Table A-5 The five-layer CNN architecture.

| Layer | Size |
|---|---|
| Input | 28x28x1 |
| Convolution + ReLU | 3x3x32 |
| Convolution + ReLU | 3x3x64 |
| Max Pooling | 2x2 |
| Dropout | 0.5 |
| Fully Connected + ReLU | 128 |
| Dropout | 0.5 |
| Fully Connected | 10 |

## 10  DETAILS OF MOTIVATIONS

### 10.1  MOTIVATING LAYER-WISE ATTACKS

We introduce a constraint loss attack inspired by the constraint module from 3DFed (Li et al., 2023) to evade distance-based defenses. We illustrate the challenges in bypassing distance-based defenses with model space attacks, thus motivating the necessity of fine-grained layer-wise attacks. The constraint loss attack circumvents distance-based defenses by incorporating a distance regularization term in the training loss function, as shown below:

$$f_{malicious} = \beta \times \frac{1}{|D_{poison}^{(i)}|} \sum_{(x,y) \in D_{poison}^{(i)}} \ell(x, y; \boldsymbol{w}_{malicious}^{(i)}) \tag{5}$$
$$+ (1 - \beta) \times \left\| \boldsymbol{w}_{malicious}^{(i)} - \boldsymbol{w}_{global}^{(i)} \right\|_2,$$

where $\boldsymbol{w}_{global}^{(i)}$ is the global model aggregated from previous rounds and $\beta \in [0, 1]$ is a hyperparameter that controls the constraint level. Fig. A-8 shows that the constraint loss attack fails to bypass the detection of MultiKrum and FLARE even setting $\beta = 0.001$, which calls for a refiner backdoor attack poisoning in layer space instead of poisoning in model space. Thus, we further explore backdoor-critical (BC) layers for refining backdoor attacks in §3.

### 10.2  MOTIVATING ATTACKS FOR SIGN-BASED DEFENSE

Unlike distance-based and inversion-based defense methods, sign-based defense methods (Bernstein et al., 2018; Ozdayi et al., 2021) measure the changes of individual update parameters—malicious

Table A-6 Main task accuracy and BSR on IID dataset. We mark the BSR below 10% (corresponding to ten classes of the datasets) as red, indicating a failed attack, and mark the highest BSR as **bold** within the same setting. The Baseline is BadNets (Gu et al., 2017). The results are the average of five repeated experiments. For LP attack (LF attack), $a\pm b$, where $a$ is the mean value, and $b$ is the standard deviation. Acc: main task accuracy (%), Avg: average, BSR unit: %.

| Model (Dataset) | | VGG19 (CIFAR-10) | | | ResNet18 (CIFAR-10) | | | CNN (Fashion-MNIST) | | |
|---|---|---|---|---|---|---|---|---|---|---|
| Attack | | Baseline | LP Attack (LF Attack) | DBA | Baseline | LP Attack (LF Attack) | DBA | Baseline | LP Attack (LF Attack) | DBA |
| FedAvg (IID) | Best BSR | 79.27 | **94.89**±0.33 | 52.9 | 76.76 | **95.94**±0.32 | 22.54 | **100** | 87.21±3.53 | 99.99 |
| | Avg BSR | 73.13 | **93.76**±0.35 | 40.67 | 70.2 | **95.35**±0.32 | 16.93 | **99.97** | 80.81±5.79 | 99.92 |
| | Acc | 83.86 | **84.16**±0.09 | 83.82 | 80.38 | **80.27**±0.64 | 80.37 | 89.05 | **89.73**±0.12 | 89.22 |
| FLTrust (IID) | Best BSR | 93.45 | **96.01**±1.35 | 75.28 | **95.97** | 95.66±1.09 | 66.52 | 99.14 | 95.84±0.2 | **100.0** |
| | Avg BSR | 85.93 | **85.99**±5.99 | 45.93 | 67.1 | 56.93±16.07 | 23.39 | 98.47 | 91.35±1.4 | **100.0** |
| | Acc | **83.26** | 82.46±0.47 | 82.94 | 74.97 | 70.77±5.4 | 77.72 | 90.0 | **90.68**±0.17 | 90.22 |
| FLAME (IID) | Best BSR | 5.26 | **91.85**±1.08 | 5.09 | 4.79 | **92.41**±0.26 | 6.08 | 0.8 | **92.37**±1.32 | 0.22 |
| | Avg BSR | 2.58 | **82.93**±1.91 | 2.16 | 3.33 | **89.41**±1.22 | 3.68 | 0.53 | **90.72**±1.03 | 0.16 |
| | Acc | **79.86** | 75.42±0.44 | 79.69 | **78.77** | 75.85±0.44 | 78.46 | 88.76 | 88.85±0.42 | **88.9** |
| RLR (IID) | Best BSR | 84.59 | **94.87**±0.15 (1.12±0.18) | 44.22 | 84.62 | **90.16**±3.39 (2.08±0.57) | 23.41 | 8.44 | 0.0 (**44.5**±2.75) | 4.99 |
| | Avg BSR | 77.46 | **89.43**±3.28 (0.65±0.12) | 34.84 | 66.32 | **72.29**±6.07 (0.96±0.26) | 9.51 | 5.38 | 0.0 (**40.85**±4.74) | 2.4 |
| | Acc | 80.7 | 79.07±0.27 (**80.78**±0.16) | 80.55 | **77.57** | 76.88±0.73 (0.96±0.17) | 77.19 | 86.86 | 87.36±0.19 (**87.69**±0.08) | 87.04 |
| MultiKrum (IID) | Best BSR | 6.9 | **96.23**±1.09 | 4.93 | 6.34 | **96.18**±0.35 | 5.16 | 0.18 | **88.22**±9.83 | 2.81 |
| | Avg BSR | 2.54 | **94.41**±0.28 | 2.42 | 3.19 | **95.24**±0.3 | 2.94 | 0.08 | **82.5**±7.27 | 1.57 |
| | Acc | **81.12** | 80.33±0.23 | 81.43 | **79.89** | 78.19±0.69 | 78.4 | **88.64** | 88.56±0.04 | 88.46 |
| FLDetector (IID) | Best BSR | 5.46 | **94.46**±1.51 | 6.56 | 4.89 | **97.68**±0.4 | 7.2 | 0.03 | 98.38±0.47 | **100.0** |
| | Avg BSR | 5.43 | **92.65**±1.01 | 6.51 | 4.43 | **97.65**±0.42 | 4.07 | 0.03 | 98.24±0.53 | **100.0** |
| | Acc | 62.86 | 52.17±3.34 | **63.3** | 68.54 | **65.02**±1.35 | 69.02 | **80.56** | 80.56 ±0.12 | 80.06 |
| FLARE (IID) | Best BSR | 4.33 | **83.12**±5.5 | 4.57 | 6.18 | **92.37**±1.07 | 5.21 | 0.09 | 90.32±1.88 | **100.0** |
| | Avg BSR | 2.33 | **68.78**±9.3 | 2.94 | 3.76 | **88.19**±1.84 | 3.29 | 0.05 | 87.97±0.83 | **100.0** |
| | Acc | **83.75** | 82.32±1.01 | 83.55 | 77.4 | **76.56**±0.56 | 75.94 | **89.19** | 89.04±0.13 | 88.98 |

Table A-7 Hyper-parameter settings.

| | Description | Fashion-MNIST | CIFAR-10 |
|---|---|---|---|
| $N$ | # of clients | 100 | |
| $C$ | Selected clients proportion | 10% | |
| $E$ | Local epoches | 2 | |
| $B$ | Local batch size | 64 | |
| $R$ | Global model training rounds | 200 | |
| $M/N$ | Malicious client proportion | 10% | |
| $PDR$ | Malicious data proportion | 50% | |
| $lr$ | Local learning rate | 0.01 | 0.1 |

clients' update parameters usually change in an opposite direction of benign clients'—leading to a divergence in the signs of parameters. Intuitively, this sign-based defense strategy works because the FL server can detect conflicts between the parameter signs of malicious and benign clients' updates and then defend against attacks by reversing the signs of learning rates to malicious update parameters.

However, models from different clients with parameters of non-consensus signs are common in FL since models are trained on clients' private data, respectively. To substantiate this observation, we conduct experiments on a CNN trained on an IID fashion-MNIST dataset, excluding any malicious clients. As shown in Fig. A-10 , our experiments demonstrate that RLR flips a significant number of learning rates of dimensions in the CNN even in the absence of any malicious clients. This vulnerability in RLR motivates us to develop a flipping attack to exploit its weaknesses.

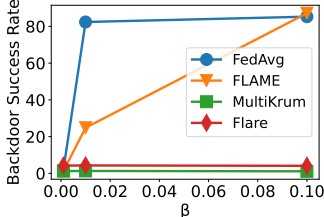

Figure A-8 Backdoor success rate (BSR) of constraint loss (CL) attack under FedAvg, FLAME, MultiKrum with varying $\beta$ values. Experiments train ResNet18 on IID CIFAR-10 dataset.

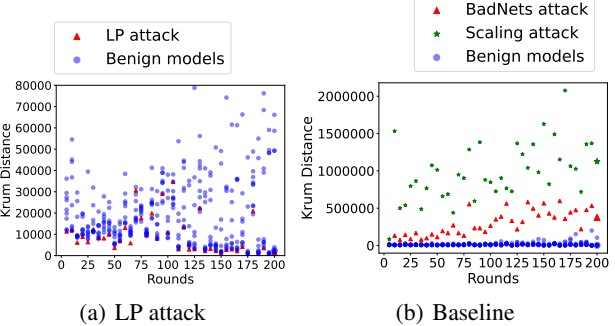

(a) LP attack       (b) Baseline

Figure A-9 Distance scores of malicious and benign model updates (red points and green points indicate malicious models, and blue points indicate benign models). It should be noted that we plot the distance scores every five rounds, *i.e.*, Round 0, Round 5, and so on, for a clear presentation.

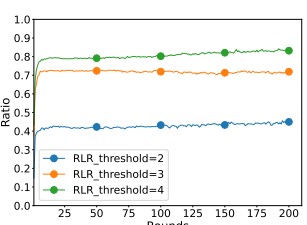

Figure A-10 Ratio of reversed dimensions in CNN models with varying RLR threshold values. Experiments train CNN on IID Fashion-MNIST dataset without malicious clients.

## 11 DETAILS OF EXPERIMENT SETUP

### 11.1 FL SYSTEM SETTINGS

The proportion of clients selected in each round among $n = 100$ clients is $C = 0.1$. Each selected clients train $E = 2$ epochs in the local dataset with batch size $B = 64$. The server trains the global model with $R = 200$ rounds to make it converge. We set $\tau = 0.95$ when identifying the BC layers via Layer Substitution Analysis (refer to Fig. 4 for sensitivity analysis of $\tau$). We set $\lambda = 1$ when training on CIFAR-10 and $\lambda = 0.5$ when training on Fashion-MNIST (refer to Fig. 5 for sensitivity analysis of $\lambda$). As for sampling malicious clients, following previous works (Sun et al., 2019), we consider fixed frequency attack and set the frequency to $M/N \times C \times n = 1$. Table A-7 in Appendix shows the detailed hyperparameter settings.

**Datasets:** Fashion-MNIST (60,000 images for training and 10,000 for testing with ten classes) and CIFAR-10 (50,000 for training and 10,000 for testing with ten classes).

**Data distribution:** Following Cao et al. (2021); Fang et al. (2020), we use $q = 0.5$ by default. Following previous works (Cao et al., 2021; Fang et al., 2020), we create non-IID datasets by dividing clients into $X$ groups according to the $X$ classes of the dataset. The possibility of samples with label $x$ assigned to the $x$-th group is $q$, while the possibility of being assigned to other classes is $\frac{1-q}{X-1}$. Samples in the same group are distributed to clients uniformly. A larger $q$ value means a higher degree of non-IID. Our experiments use $q = 0.5$ by default. Our sensitivity analysis experiments run on IID datasets since defense strategies are more likely to detect malicious models as outliers in IID setting, which further justifies the effectiveness of attack methods.

**Models:** We use the following three models: A five-layer CNN (Cao et al., 2021; Zhang et al., 2022b), ResNet18 (He et al., 2016), and VGG19 (Simonyan & Zisserman, 2014). The CNN model is trained on Fashion-MNIST, while ResNet18 and VGG19 are trained on CIFAR-10 (the detailed CNN structure refers to Table A-5 in Appendix).

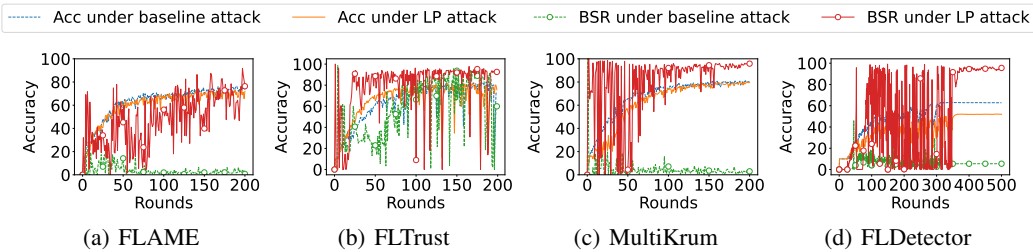

Figure A-11 VGG19 trained with robust different aggregation rules on IID data. Acc indicates the main task accuracy, and BSR indicates Backdoor Success Rate.

## 11.2 DEFENSE METHODS & SETTINGS

***FLTrust*** (Cao et al., 2021): We enlarge the size of the root dataset from 100 in the original paper to 300, which enables the server to detect attacks precisely.

***RLR*** (Ozdayi et al., 2021): We reuse and set the threshold of learning rate flipping in each parameter to 4 as same as the original paper, where RLR claims that the threshold should be larger than the number of malicious clients (Our experiments have one malicious client in each round).

***FLAME*** (Nguyen et al., 2021): `min_cluster_size` is set to $n/2 + 1$, `min_samples` to 1, and the noise parameter to 0.001 following the original paper.

***MultiKrum*** (Blanchard et al., 2017): The server calculates squared distance called Krum distance through the closest $N \times C - f$ clients updates, where $f$ is a hyperparameter and assumed to be equal or larger than the number of malicious clients, which is 1 in our setting. In our experiments, four clients with the highest distance score are selected by MultiKrum to aggregate the global model with $f = 2$.

***FLDetector*** (Zhang et al., 2022b): We reuse the same setting in the original paper. All clients perform a single step of standard gradient descent and submit their corresponding model updates to the server in each round. As a result, the fraction of clients participating in the aggregation is set to $C = 1$, the local epoch is set to $E = 1$, and the number of training rounds is enlarged to $R = 500$. Additionally, the window size is set to 10 and attacks start after the server finishes the initialization following the original paper.

***Flip*** (Zhang et al., 2022a): We adopt the confidence threshold of 0.4 used in the original paper. The effectiveness of adversarial training might be impeded due to the inability of trigger inversion to invert the same trigger used by the attacker, as reported in the literature. We assume that the server has prior knowledge of the shape of the trigger, which helps to successfully unlearn the backdoor task.

***FLARE*** (Wang et al., 2022): We reuse the setting in the original paper, where the root dataset comprises 10 samples per class.

## 11.3 BASELINE ATTACKS

**BadNets** (Gu et al., 2017) injects a $5 \times 5$ square as triggers into the bottom right corner of the images and relabel those images as a targeted label on each malicious client. In our experiments, we set the targeted label to Class 5 of the Fashion-MNIST and CIFAR-10 datasets. BadNets is a typical attack, which is regarded as a baseline attack in the following experiments by default. **DBA** (Xie et al., 2019) splits the whole $5 \times 5$ trigger pattern into four smaller triggers ($2 \times 2$, $2 \times 3$, $3 \times 2$, and $3 \times 3$). **Subnet Replacement Attack (SRA)** (Qi et al., 2022) uses the same subnet structure in ResNet18 from the original paper. **Scaling attack** (Bagdasaryan et al., 2020) scales up parameters to implement model replacement, where a scale factor is set to 5 in our experiments. According to the original paper, Scaling attack should start after the global model is close to convergence.

Table A-8 Subnet Replacement Attack (SRA) in FL attacks ResNet18 trained on CIFAR-10 (Acc: the main task accuracy).

| Attack | FedAvg | | | MultiKrum | | | FLAME | | |
|---|---|---|---|---|---|---|---|---|---|
| | Best BSR (%) | Avg BSR (%) | Acc (%) | Best BSR (%) | Avg BSR (%) | Acc (%) | Best BSR (%) | Avg BSR (%) | Acc (%) |
| LP Attack (IID) | 96.63 | 95.38 | 80.44 | **95.91** | **95.21** | 79.17 | **98.51** | **98.17** | 77.21 |
| SRA (IID) | **98.14** | **97.92** | 79.84 | 4.44 | 2.61 | 80.53 | 6.26 | 4.36 | 76.72 |
| LP Attack (non-IID) | 94.52 | 87.34 | 76.67 | **96.2** | **93.64** | 76.65 | **91.43** | **89.51** | 71.45 |
| SRA (non-IID) | **98.11** | **96.66** | 78.68 | 8.46 | 3.52 | 73.3 | 17.66 | 4.81 | 74.4 |

Table A-9 Malicious clients acceptance rate (MAR) and Backdoor Success Rate (BSR) of constrain loss (CL) attack on ResNet18, trained on the IID CIFAR-10 dataset, for varying values of $\beta$. The MAR is not applicable for FedAvg since it does not filter out malicious clients. BSR unit: %, MAR unit: %.

| Attack | FedAvg | FLARE | MultiKrum | | FLAME | |
|---|---|---|---|---|---|---|
| | BSR | BSR | MAR | BSR | MAR | BSR |
| LP Attack | **95.3** | **88.19** | **94.2** | **95.1** | **99.0** | 89.9 |
| CL Attack $\beta$=0.1 | 92.7 | 3.29 | 0 | 3.15 | 66.2 | **94.1** |
| CL Attack $\beta$=0.01 | 83.4 | 4.1 | 0 | 3.2 | 78.3 | 28.2 |
| CL Attack $\beta$=0.001 | 3.8 | 4.03 | 0 | 3.1 | 84.5 | 3.8 |

## 12 CASE STUDY

### 12.1 COMPARISON WITH SRA (QI ET AL., 2022)

Designed for the deployment stage, SRA does not require knowledge of the model parameters and only necessitates the alteration of a limited number of parameters to ensure the preservation of main task performance. Though not specifically designed for FL, SRA can compromise the classical FL process (*e.g.*, FedAvg) with only 10% malicious clients. However, despite the relatively small modification of 3% of the model parameters, the crafted models produced by the SRA exhibit significant differences from benign models, making them susceptible to be detected by distance-based defense mechanisms such as FLAME and MultiKrum. Table A-8 shows that SRA achieves over 97% average BSR against FedAvg but less than 5% average BSR against MulitKrum and FLAME.

### 12.2 COMPARISON WITH CONSTRAIN LOSS ATTACK

Table A-9 presents the results of the evaluation of constrain loss attack with respect to its ability to decrease the cosine similarity with benign models and bypass the detection mechanism of FLAME. The results indicate that when $\beta$ is set to 0.1 or 0.01, constrain loss attack is able to attack FLAME defense leading BSR to 94% even higher than LP attack (89.9%) by reducing the cosine similarity with benign models. However, the performance of LP attack remains superior to Constrain Loss attack under MultiKrum and Avg, where LP attack achieves both 95% in BSR higher than 3% and 92% in constrain loss attack. As the value of $\beta$ decreases, the constraint imposed by the distance to benign models becomes stricter. Consequently, the effectiveness of constrain loss attack is reduced and its performance deteriorates when $\beta$ equals 0.1 or 0.01, which is shown in FedAvg and FLAME settings. While constrain loss attack sacrifices the effectiveness of attack to hide the distance deviations by decreasing $\beta$ to 0.001, our results show that MultiKrum and FLARE are able to detect constrain loss attack in all settings of $\beta$ (Table A-9 ), whereas our LP attack can evade and corrupt MultiKrum and FLARE.

### 12.3 LP ATTACK AND SCALING ATTACK AGAINST FLIP DEFENSE (ZHANG ET AL., 2022A)

The experimental results, presented in Fig. A-12 , reveal that Flip defense can reduce the BSR of BadNets attack and LP attack in the normal setting $\lambda = 1$. However, we also observe that Flip defense fails to address the threat of model replacement from Scaling attack, which can overwrite the global model by scaling up the parameters in the malicious models. As a consequence, LP attack

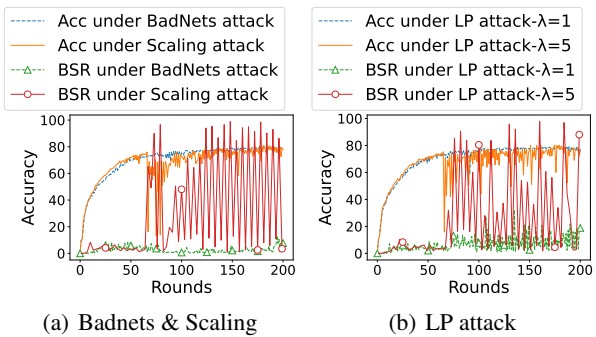

(a) Badnets & Scaling     (b) LP attack

Figure A-12 LP attack, BadNets attack, and Scaling attack against Flip (Zhang et al., 2022a).

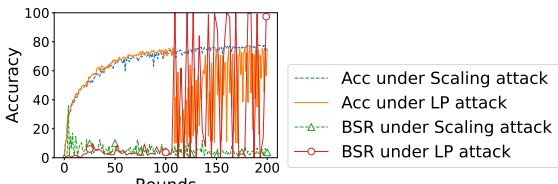

Figure A-13 LP attack ($\lambda = 5$) and Scaling attack against the combination of Flip (Zhang et al., 2022a) and MultiKrum.

can also corrupt the Flip defense and maintain the main task accuracy at a high level by setting $\lambda = 5$, which can be regarded as a combination of LP attack and Scaling attack. The fluctuations in BSR are caused by the fact that the malicious models perform the model replacement only when the global model has no backdoor task injected. When the global model learns the backdoor task, adversarial training can help the global model unlearn the backdoor task and significantly reduce the BSR. However, when the BSR is low and the loss of backdoor task is large, malicious models perform model replacement again, leading to high BSR and fluctuations. Naturally, the defender can combine Flip with MultiKrum to mitigate Scaling attack. Fig. A-13 shows that the combination of Flip and MultiKrum can adaptively impede Scaling attack but LP attack can bypass the detection and inject backdoor into the global model.

# 13   ROBUSTNESS OF LAYER-WISE POISONING ATTACK

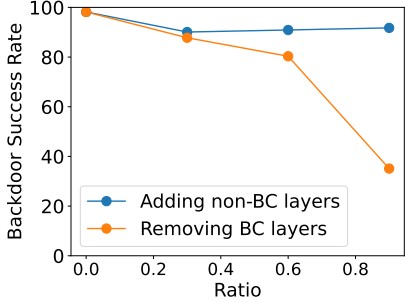

Figure A-14 Robustness of LP attack. A specific ratio of layers identified as BC layers are randomly removed or a number of layers identified as non-BC layers (equal to a specific ration of the BC layers) are added.

To evaluate the robustness of LP attack, we simulate the scenario where the identification of BC layers may be inaccurate, potentially missing a portion of BC layers or including non-BC layers. We conduct an experiment that randomly removes a specific ratio of BC layers in $L^*$ and an experiment that randomly adds a number of non-BC layers into $L^*$. Fig. A-14 demonstrates that LP attack is able to maintain a high BSR even when 60% of the BC layers are removed or when a significant number of non-BC layers (equal to 90% of the BC layers) are added.

| Attack | Defense | BSR (%) | Acc (%) |
|--------|---------|---------|---------|
| Baseline | FLAME | 1.5 | 83.6 |
| LP Attack | FLAME | 90.9 | 83.6 |
| Baseline | MultiKrum | 2.0 | 83.1 |
| LP Attack | FLAME | 97.2 | 80.1 |

Table A-10 LP Attack on FEMNIST.

Table A-11 The performance of adaptive defense under LP attack. Experiments are conducted on the CIFAR-10 dataset.

| Distribution | Model | BSR (%) | Acc (%) |
|--------------|-------|---------|---------|
| IID | ResNet18 | 87.32 | 75.79 |
|  | VGG19 | 94.8 | 80.22 |
| non-IID | ResNet18 | 52.39 | 69.33 |
|  | VGG19 | 64.33 | 69.82 |

## 14 RESILIENCE AGAINST LAYER-WISE DEFENSE

When a defense strategy realizes that the LP attack only poisons specific BC layers, it may perform adaptive layer-wise detection and augment the defense for those specific layers. To demonstrate the resilience of LP attack against such a layer-wise defense strategy, we design an adaptive defense and evaluate its impact on the LP attack. According to Table 1, MultiKrum is the most effective defense for the LP attack. So we extend it to a layer-wise MultiKrum.

Let $\boldsymbol{u}_{t+1} = [\boldsymbol{u}_{t+1,1}, \boldsymbol{u}_{t+1,2}, \dots, \boldsymbol{u}_{t+1,l}]$ denotes the global model $\boldsymbol{w}_{t+1}$ in round $t+1$, where $\boldsymbol{u}_{t+1,j}$ is the $j$-th layer. We use MultiKrum (Blanchard et al., 2017) to aggregate $j$-th layer's parameters as:

$$\boldsymbol{u}_{t+1,j} = \texttt{MultiKrum}(\boldsymbol{u}_{t,j}^{(1)}, \boldsymbol{u}_{t,j}^{(2)}, \dots, \boldsymbol{u}_{t,j}^{(N)}), \tag{6}$$

where the input $\boldsymbol{u}_{t,j}^{(i)}$ in function $\texttt{MultiKrum}(\cdot)$ is a vector of $j$-th layer in client $i$. The proposed layer-wise MultiKrum chooses the $N \times C - f$ closest vectors of the layer as benign layers and aggregates those layers for the new layer in $(t + 1)$-th round global model $\boldsymbol{w}_{t+1}$. Here, $N \times C$ is the number of clients selected in each round and $f$ is the hyperparameter in MultiKrum algorithm. We extend MultiKrum to layer-wise MultiKrum safely aggregating parameters in each layer.

Table A-11 shows that layer-wise MultiKrum cannot detect and filter LP attack effectively. The BSR is up to 94% in VGG19 in IID setting. We calculate krum distance in each BC layer. Fig. A-15 shows that though the BC layer—`linear.weight` is easily distinguished by Krum distance, other BC layer—convolutional layers present limited deviation from benign layers. Those BC layers carry backdoor tasks into the global model successfully bypassing layer-wise MultiKrum detection.

## 15 IMPACT OF RATIOS OF MALICIOUS CLIENTS

While LP attack performs well with $M/N = 0.1$, we further explore if it can work with extremely low ratios of malicious clients. In Fig. A-16 , we conduct experiments on ResNet18 trained on non-IID CIFAR-10 datasets under FLAME and MultiKrum. The results indicate our LP attack still works

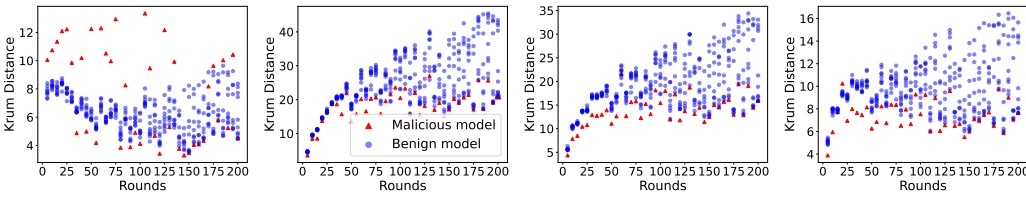

(a) linear.weight    (b) layer3.0.conv2.weight  (c) layer4.1.conv1.weight  (d) layer4.1.conv2.weight

Figure A-15 Krum distance of BC layers in LP attack. The experiment is conducted on ResNet18 trained on IID CIFAR-10 dataset.

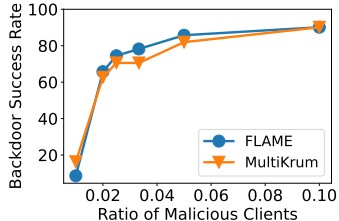

Figure A-16 Performance of LP attack with different ratios of malicious clients.

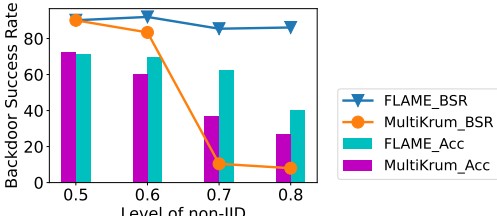

Figure A-17 Performance of LP attack under different degrees of non-IID data distribution.

well with an extremely low ratio of malicious clients $M/N = 0.02$, where the attacker implements one backdoor attack in each of five rounds. However, our LP attack does not attack successfully with $M/N = 0.01$, where the attacker injects backdoor in every ten rounds. The possible reason is that backdoor attack is neutralized by the average step in the server.

## 16 IMPACT OF NON-IID DATA

**High Level of Non-IID Data Distribution.** In previous sections, we have shown that LP attack works well on the non-IID dataset with $q = 0.5$. To explore if higher degrees of non-IID affect the identification of BC layers, we conduct experiments on ResNet18 trained with the CIFAR-10 dataset under FLAME and MultiKrum. The results show that our LP attack can attack FLAME in a high level of non-IID distribution settings $q = 0.8$. However, LP attack loses its effectiveness facing MultiKrum when $q = 0.7$ and $q = 0.8$, where the model might not learn well on neither main task nor the backdoor task (the main task accuracy is lower than 40%).

**Experiments on Real-world Non-IID Datasets.** FEMNIST is a real-world dataset included in LEAF (Caldas et al., 2018). FEMNIST dataset comprises 805,263 images, which are distributed into 3550 devices. In our experiment, we group all devices into 100 clients and set the epoch to 1. Results on Table A-10 demonstrate that the LP Attack successfully injects a backdoor, whereas the baseline attack proves unsuccessful.

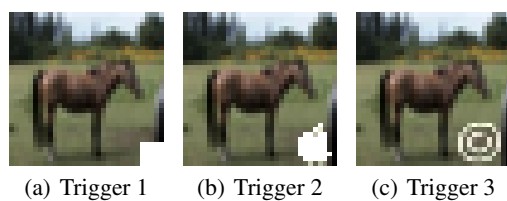

(a) Trigger 1    (b) Trigger 2    (c) Trigger 3

Figure A-18 Different backdoor trigger shapes in an CIFAR-10 data sample ("square," "apple," and "watermark").

Table A-12 BC layer count under varying learning rates (lr) for overfitting training.

| | epoch | 20 | 30 | 40 | 50 |
|---|---|---|---|---|---|
| | 0.01 | 7 | 9 | 9 | 10 |
| **lr** | 0.05 | 5 | 5 | 6 | 5 |
| | 0.1 | 5 | 4 | 4 | 5 |

## 17 MORE ANALYSIS ON BC LAYERS IN CV

### 17.1 INFLUENCE OF HYPERPARAMETERS

**Shapes of Backdoor Triggers**. We explore the influence of different trigger shapes for BC layer identification. Fig. A-18 presents three different trigger shapes applied to backdoor a ResNet18 model, trained on the CIFAR-10 dataset. Fig. A-19 shows that different triggers may lead to different BC layer identifications, but `linear.weight` is constantly identified as a BC layer for all three shapes.

**Types of Backdoor Attacks**. In addition to BadNets attacks, we also investigate the presence of BC layers in other types of backdoor attacks, such as GAN-based dynamic backdoor attacks Salem et al. (2022) and filter-based backdoor attacks Cheng et al. (2021). In the case of dynamic triggers Salem et al. (2022), our analysis reveals seventeen BC layers (41%) on ResNet18 trained on the CIFAR-10 dataset, a significant increase compared to pattern triggers (9%). In contrast, for Instagram filter triggers Cheng et al. (2021), only three convolutional layers (7%) are identified as BC layers, and `linear.weight` is not considered as BC layers, unlike the consistent identification with pattern triggers. The detail of BC layers refers to Fig. A-20 . Our findings demonstrate that our BC layer detection approach is versatile and applicable to various trigger types, revealing the presence of BC layers with different types of backdoor attacks.

**Model Initial Weights**. We study the identification of BC layers in a model with different initial weights. We conduct repeated identification on the ResNet18 trained on CIFAR-10 dataset. Fig. A-21 shows that if a model is initialized with different parameters, the identification of BC layers is likely the same, indicating that changing model parameters does not lead to different BC layers.

**Trainig Hyperparameters**. We explore the impact of training hyperparameters on BC layers in ResNet18 trained on CIFAR-10 dataset. The results in Table A-12 indicates that the smaller learning rate and the larger training epochs cause the increase in the number of BC layers. But the number of BC layers in the entire model is relatively low, with only 10 out of 62 layers being BC. It demonstrates the existence of BC layers.

### 17.2 ANALYSIS OF BC LAYERS IN CV MODELS

In this section, we conduct BC layer analysis in more CV models especially large models. Then we further verify that the existence and identification of BC layers are consistent under different locations and shapes of backdoor triggers, different model initial weights, and different datasets.

**BC Layers in Large Models.** Fig. A-22 shows that a small subset of layers, referred to as BC layers, can be observed in different architectures of neural networks on CIFAR-10 and CIFAR-100 datasets, respectively. Our findings reveal that these BC layers primarily consist of a small ratio of weight layers (*e.g.*, `fc1.weight` layer) and *no* bias layers (*e.g.*, `fc1.bias` layer). Furthermore, we observe that deeper layers are more likely to be selected as BC layers, specifically the `linear.weight` layer as the last layer, which has been consistently identified as a BC layer in ResNet18 (He et al., 2016), as Fig. A-19 and Fig. A-21 shown in Appendix 17. In addition, we find that BC layers in DenseNet (Huang et al., 2017), EffNet (Freeman et al., 2018), and ResNet (He et al., 2016) are mainly focusing on the weight of convolutional layers combined with one fully connected layers.

**Locations of Backdoor Triggers**. We embed a trigger to different locations shown in Fig. A-24 within a Fashion-MNIST data sample and investigate their influences over the backdoor tasks. The experiment follows the same setting in §3.2 and trains CNN (the structure of CNN refer to Table A-5 ) on Fashion-MNIST. Fig. A-23 shows that the identification of BC layers remains consistent even when the trigger's location changes within the data sample.

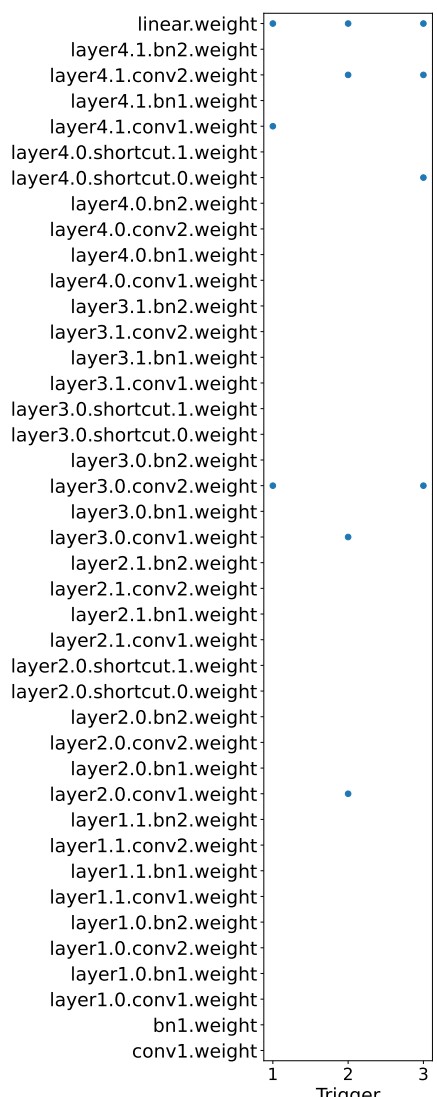

Figure A-19 A trigger's different shapes and the BC layers in ResNet18, trained on CIFAR-10.

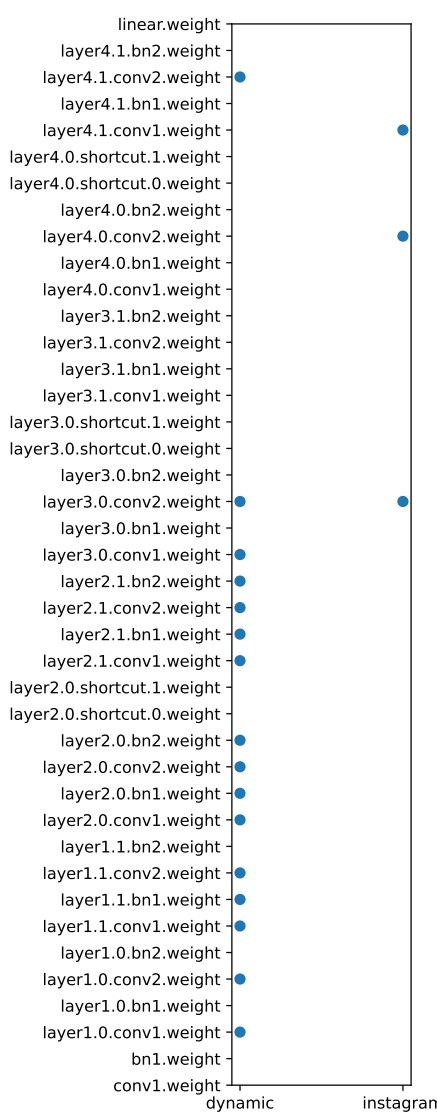

Figure A-20 BC layers on ResNet18 embedded with the dynamic trigger and Instagram filter trigger, trained on CIFAR-10.

More analysis about the shapes of triggers, types of backdoor attacks, training hyperparameters, and model initial weights refer to Appendix 17. In summary, BC layers are prevalent in diverse backdoor attacks and general CV models, and several factors collectively influence the selection of these layers. Through our experiments, we have shown that model architecture, depth, training datasets, trigger shape, and the type of backdoor attacks all contribute to varying choices of BC layers. The determination of which layers become BC is highly dependent on specific cases and scenarios.

## 17.3   ANALYSIS OF BC LAYERS IN NLP

The identification of BC layers is applicable to NLP models, which have multiple layers. We train a 2-layer LSTM as a word predictor following Bagdasaryan et al. (2020) on Reddit dataset (November

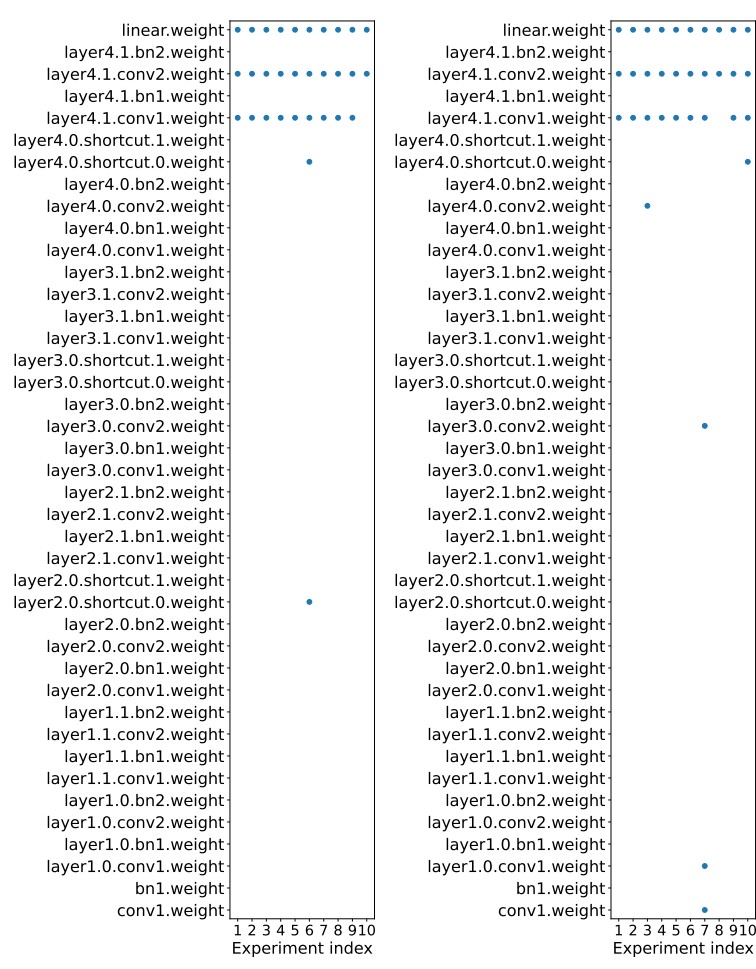

(a) Same initial model weight.    (b) Different initial model weights.

Figure A-21 Repeated layer substitution analysis on the same model initialization (Here we omit the layers that do not contribute gradients and "bias" layers since these layers are never identified as BC layers).

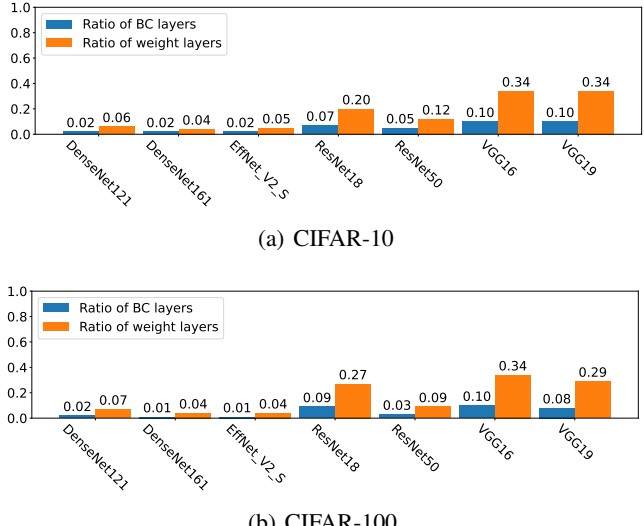

Figure A-22 The BC layers ratios in large models trained on CIFAR10 and CIFAR100. The ratio of BC layers indicates the ratios of the number of BC layers on the number of layers in models and the ratio of weight layers indicates the ratio of BC weight layers on the number of weight layers in models.

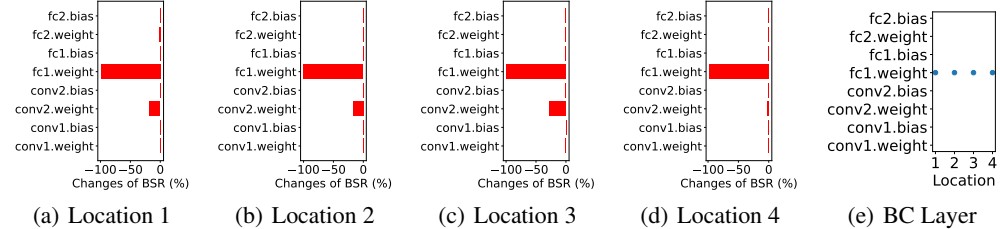

Figure A-23 The BC layers of the four-layer CNN model with a trigger at different locations.

2017),[1] and train BERT and DistilBERT as classifiers in sst2 (Socher et al., 2013) dataset following Shen et al. (2021). Fig. A-25 shows that the ratio of BC layers is less than a quarter in all layers and about a half in weight layers. We find that BC layers are only attached in the encoder layer in the 2-layer LSTM. Regarding classification tasks in NLP, BC layers lie on Q, K, and V vectors in layers closer to the output layer and linear layer of various transformer blocks. In contrast to CV classification models, the final linear layer is not BC layer in both BERT and DistilBERT.

## 18  MORE RELATED WORKS

**Backdoor Attack in Deployment Stage**. The deployment stage of a backdoor attack aims to misclassify a set of crafted samples into a targeted label by modifying a small number of model weights. Several studies, such as Bai et al. (2020); Rakin et al. (2019; 2020; 2021), have considered the manipulation of binary-form parameters stored in computer memory to inject backdoors into models. Other studies, such as Qi et al. (2022); Li et al. (2021); Tang et al. (2020), have proposed modifying a subnet of the model to identify triggers. Li et al. (2021); Tang et al. (2020) need to modify the model architecture and program, while Qi et al. (2022) selects a path from the input layer to the output layer to craft a subnet without modifying model architecture and knowing the parameters or training data in models. Despite these approaches having the same goal as us that only changing a limited number of parameters or bits, they have been found to be susceptible to be detected by FL defense strategies, as they do not take the distance as a part of the optimization.

**Backdoor Critical Parameters**. Recent research has demonstrated that certain neurons twithin neural networks are particularly susceptible to backdoor attacks. Several studies, including Wu & Wang (2021); Li et al. (2020); Wang et al. (2019); Liu et al. (2019; 2018), have proposed pruning

---

[1]https://bigquery.cloud.google.com/dataset/fh-bigquery:reddit_comments

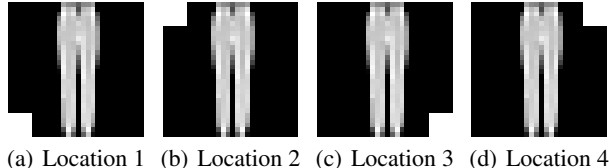

(a) Location 1   (b) Location 2   (c) Location 3   (d) Location 4

Figure A-24 Backdoor trigger locations within a data sample.

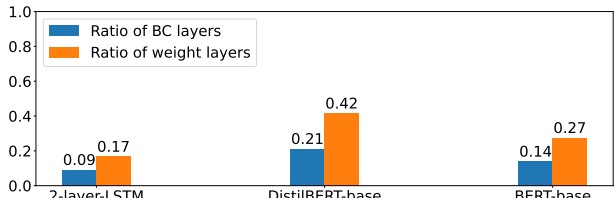

Figure A-25 The BC layers ratios in large NLP models.

these critical neurons as ap means of mitigating such attacks. Adversarial Neuron Perturbation (Wu & Wang, 2021) is a gradient-based technique that injects backdoors with a limited perturbation budget, however, it is computationally expensive. Yao et al. (2019) proposes a method for injecting backdoors into transfer learning by targeting shallow layers of the network.

**Backdoor Attack in Federated Learning.** LIE (Baruch et al., 2019) clips models updates in each parameter by calculating the mean and variance, but this attack fails in classical setting FedAvg and new defense strategies like FLAME and FLDetector with a relatively large number of attackers (20%). Scaling attack (Bagdasaryan et al., 2020) scales up malicious clients weights to overcome the affect of other clients, which is easily detected by defense strategies. *Edge-case* backdoor attack (Wang et al., 2020) aims to attack a set of samples with low predicted probability, which is hard to detect. But this attack has disadvantages in that targeted inputs are decided by the model and it is also defended by SOTA defense like Nguyen et al. (2021); Rieger et al. (2022). 3DFed (Li et al., 2023) proposes a constraint loss module for distance-based defenses, a noise masks module for bypassing update energies detection, a decoy model module for deceiving dimensionality reduction techniques, and an indicator module for fine-tuning hyperparameters. However, the indicator module and decoy model module require multiple clients to collaborate in attacks, which conflicts with our experimental setup, where only one malicious client is allowed in each round. Consequently, these modules are not considered in the experiment section.

Durability measures the number of rounds that backdoor attacks remain in the global after attacks. Neurotoxin (Zhang et al., 2022c) injects backdoor into neurons with the smallest L2 norms to avoid the mitigation from benign clients. PerDoor (Alam et al., 2023) targets parameters that deviate less to keep adversarial samples durable.

**Model compression and lottery tickets**. Like iterative model pruning techniques widely applied to identify lottery tickets (Denil et al., 2013; Cheng et al., 2015; Frankle & Carbin, 2019), we can also iteratively prune a DNN model on poisoned data to seek for a key subnetwork that dominates its vulnerability. However, iterative model pruning requires large volumes of training data, and the identified structures (*i.e.*, the lottery ticket) vary with different initial model weights. Instead, this paper proposes a general in-situ approach that directly searches BC layers. Zhang et al. (2023) proposed FedIT to fine-tune large language models (LLMs) via a small and trainable adapter (*e.g.*, LoRA (Hu et al., 2021)) on each client. We plan to explore the backdoor-critical structures in LLM adapters in our future work.

