# OpenReview forum: "Backdoor Federated Learning by Poisoning Backdoor-Critical Layers"
_ICLR.cc/2024/Conference — ICLR 2024 poster_

### Official Review · Reviewer_EGpy · 2023-10-28

**Soundness:** 3 good
**Presentation:** 3 good
**Contribution:** 3 good
**Rating:** 6
**Confidence:** 4

**Summary:**

In this paper, the authors introduce two layer-specific backdoor attack methods that aligns with the framework of federated learning attackers.
To recognize backdoor-critical layers, they provide a layer substitution method including Local Training, Forward Layer Substitution and Backward Layer Substitution.
Their evaluation, conducted across different models (ResNet18, VGG-19, etc.) and datasets (CIFAR-10, Fashion-MNIST, etc.), demonstrates that the newly proposed layer-wise backdoor attack techniques surpass the performance of existing non-layer-wise methods for backdoor attacks.

**Strengths:**

1. Extensive experiment results
2. Clear presentation of the proposed method

**Weaknesses:**

1. It appears that the approach is tailored to a specific model. I'm curious about its performance when applied to other models within the Resnet and VGG families.

2. For the LP attack, the attacker must possess knowledge of the benign workers' model parameters, and for the LF attack, they need to ascertain whether the defense modifies the sign of these parameters.

3. To ensure a fair comparison, the authors may want to take into account layer-specific defenses such as norm-clipping specific to each layer and add noise to only specific layers.

4. Instead of solely replacing the BC layers after their identification, it might be more optimal for the authors to consider freezing the other layers and fine-tuning the BC layers using poisoned data.

**Questions:**

1. During the earlier stages and as FL training nears convergence, do the characteristics of the BC layer change, or is it primarily determined by the model architecture?

2. It's noticeable that all BC layers in the provided experiments are fully connected (fc) layers. Is there a specific rationale or intuition behind this? Given that BC layers are often the first fc layer, it could be intriguing to explore adjustments in their structure, size, and hyperparameters. Additionally, is there a possibility that within a single BC layer, certain parameters or neurons are more crucial than others? If this is the case, why not consider selecting critical parameters or neurons across multiple layers instead of exclusively focusing on critical layers?

3. What is the reasoning behind the sequence of Local Training, Forward Layer Substitution, and Backward Layer Substitution, as opposed to individually training each layer using poisoned data and subsequently sorting them?

---

> ### Author Response · Authors · 2023-11-16
>
> Responses to Weaknesses:
>
> W1: A: Please refer to Section 16 of the Appendix, where we conducted experiments to show that BC layers exist in various CV and NLP model architectures, including VGG16, ResNet34, EffNet, BERT, and so on.
>
> W2: A: As mentioned in Section 4.1, the attacker can utilize the data in malicious clients to train clean models to simulate benign clients [1].  Therefore, the possession of knowledge of benign clients is unnecessary for our approach.
>
> [1] Fang, Minghong, et al. "Local model poisoning attacks to {Byzantine-Robust} federated learning." 29th USENIX security symposium (USENIX Security 20). 2020.
>
> W3: A: Thanks for your suggestions. Indeed, we considered layer-specific defenses. We introduced an adaptive defense - layer-wise Multikrum in Section 13 of the Appendix. Experiments Table A-10 shows that our LP attack can be effective against this defense and achieve 95% and 64% BSR in IID and Non-IID CIFAR-10 datasets respectively when attacking VGG19.
>
> W4: A: It is a possible idea. In this paper, we train a benign model and a malicious model for layer substitution. One of the advantages of our approach is that stealthiness can be controlled by the number of attack layers, where we introduce an adaptive layer control mechanism to adjust the number of attack layers each round. We will explore this idea in our future work.
>
> Responses to Questions:
>
> Q1: A: The set of BC layers undergoes changes throughout the training process. As shown in Fig. 6, lower frequencies in BC layer identification correspond to lower BSRs, signifying alterations in BC layers over time.
>
> Q2: A: Please refer to Section 16 of the Appendix, where we explored how various factors, such as the shape of triggers, types of backdoor attacks (types of triggers), model initial weights, training hyperparameters, and model architecture, influence the behavior of BC layers. It is possible that some neurons are more crucial than others and some existing works have explored this [2] as mentioned in Section 17 of the Appendix. They utilize Backpropagation to search those neurons, which is computationally demanding. LP attack uses layer substitution analysis to search BC layers, which requires fewer computing resources. Because layer substitution utilizes more forward propagation instead of backpropagation.
>
> [2] Wu, Dongxian, and Yisen Wang. "Adversarial neuron pruning purifies backdoored deep models." Advances in Neural Information Processing Systems 34 (2021): 16913-16925.
>
> Q3: A: In our experiments, training one layer and freezing other layers may not achieve high main task accuracy and high BSR at the same time. We conduct an experiment in which we freeze other layers and fine-tune the backdoor task within a single layer. The results in the table below show that neither a fully connected layer nor a convolutional layer achieves high BSR and high Acc at the same time. From our observation, multiple layers can effectively collaborate to learn backdoors in deep learning networks like ResNet and VGG. So LP attack trains a benign model and a malicious model for the layer substitution analysis.
>
> | Training Layer        | BSR (%) | Acc (%) |
> | --------------------- | ------- | ------- |
> | linear.weight         | 34.66   | 82.74   |
> | layer4.1.conv2.weight | 47.32   | 31.61   |

---

> > ### Comment · Reviewer_EGpy · 2023-11-23
> >
> > I appreciate the authors' response and informative clarification. After reading the rebuttal and other reviewers' comments, part of my concerns has been addressed.

---

> > > ### Author Response · Authors · 2023-11-23
> > >
> > > We will be glad to discuss if anything still concerns you. Could you please consider updating the score if our rebuttal is helpful? We will continue improving the submission's quality based on your suggestions. Thank you!

---

### Official Review · Reviewer_MrXZ · 2023-10-31

**Soundness:** 3 good
**Presentation:** 2 fair
**Contribution:** 2 fair
**Rating:** 6
**Confidence:** 4

**Summary:**

The paper proposes a backdoor attack in Federated Learning (FL) by identifying Backdoor-Critical (BC) layers, a subset of layers crucial for model vulnerabilities. The paper verifies and exploits these layers, devising a new backdoor attack strategy that balances effectiveness and stealthiness. The introduced Layer Substitution Analysis algorithm and two layer-wise backdoor attack methods, LP and LF Attack, minimize model poisoning while successfully infiltrating FL systems, outperforming recent methods even with a lower percentage of malicious clients.

**Strengths:**

+ Layer Substitution Analysis can identify the existence of backdoor-critical layers.
+ Utilize the knowledge of backdoor-critical layers to craft effective and stealthy backdoor attacks with minimal model poisoning.
+ Demonstrates effectiveness in bypassing state-of-the-art defense methods and injecting backdoors into models with a small number of compromised clients.

**Weaknesses:**

The core concept of this paper centers around identifying critical parameters within the model to facilitate effective backdoor insertions. While the strategy of embedding persistent backdoors through carefully identifying key parameters has been previously explored in [1] and [2], the authors must outline the novelty of the proposed method when compared to the techniques presented in [1] and [2].

The paper does not provide any comparisons between the proposed method and the recent state-of-the-art backdoor insertion technique presented in [3].

The primary assumption for identifying the critical layer for backdoor insertion hinges on the global model reaching saturation. However, given the continuous nature of federated learning, the timeline to model saturation can be extensively prolonged based on dataset distributions and the application domain, thereby escalating the adversary's complexity. Furthermore, the persistent nature of the injected backdoor amidst the continuous updates in federated learning remains unclear. The authors need to provide further clarifications on this aspect.

In a paper focused on federated learning, it is expected to see some experimental evaluations conducted on the LEAF [4] benchmark dataset.

[1] Z Zhang et al., "Neurotoxin: Durable Backdoors in Federated Learning", ICML 2022.
[2] M Alam et al., "PerDoor: Persistent Backdoors in Federated Learning using Adversarial Perturbations", IEEE COINS 2023.
[3] H Li et al., "3DFed: Adaptive and Extensible Framework for Covert Backdoor Attack in Federated Learning", IEEE S&P 2023.
[4] https://leaf.cmu.edu/

**Questions:**

1. How does the proposed method diverge from the techniques presented in [1] and [2] in terms of novelty?
2. Provide comparative analysis between the proposed method and the state-of-the-art backdoor insertion technique outlined in [3].
3. How does the injected backdoor maintain its persistence amidst the continuous updates inherent in federated learning?
4. Provide experimental evaluations on the LEAF [4] benchmark dataset.

---

> ### Author Response · Authors · 2023-11-16
>
> Responses to Questions:
>
> Q1: A: Thanks for the references you provided. We will cite these papers in our revision and emphasize the difference in the related work section.
> Our objectives and contributions diverge from [1] and [2]. Our primary aim is to evade detection by stringent defenses, while Neurotoxin [1] is designed to improve the durability of backdoor without accounting for stringent defenses. In terms of contributions, we propose layer substitution analysis to identify BC layers and construct LP attacks. Neurotoxin utilizes Projected Gradient Descent (PGD) to train backdoor attacks in neurons with the smallest L2 norms, safeguarding against mitigation by benign clients. Although both our paper and [1] aim to inject backdoor via a subset of parameters, our distinct goal and novel method set us apart. PerDoor[2] utilizes noise generated by adversarial perturbations as triggers, while we consider a fixed pattern designed by the attacker as triggers. PerDoor has different settings and goals from our work.
>
> [1] Zhang, Zhengming, et al. "Neurotoxin: Durable backdoors in federated learning." International Conference on Machine Learning. PMLR, 2022.
>
> [2] Alam, Manaar, Esha Sarkar, and Michail Maniatakos. "PerDoor: Persistent Backdoors in Federated Learning using Adversarial Perturbations." 2023 IEEE International Conference on Omni-layer Intelligent Systems (COINS). IEEE, 2023.
>
> Q2: A: Table 4 in the 3DFed paper shows that 3DFed (CL-only) exhibits greater efficacy against distance-based defenses compared to the complete 3DFed, wherein other modules contribute to enlarging the distance in malicious models. Thus, we implemented 3DFed (CL-only) instead of complete 3DFed to corrupt distance-based defenses, such as FLAME and Multikrum. The introduction of the CL module was detailed in Section 9.1 of the Appendix, and further experiments on 3DFed (CL-only) were conducted in Section 11.2 of the Appendix to provide a comparative analysis. The results show that 3DFed is not enough to deceive stringent distance-based defenses, such as MultiKrum and FLARE, while LP attack can achieve high BSR.
>
> Q3: A: Durability and stealthiness represent two characteristics in the context of backdoors in FL. Durability-related work and stealthiness-related work have different objectives and settings. Durability-related works assume that the attacker should stop the attack after a certain time point, and baseline attacks should be able to insert the backdoor into the server [1]. Stealthiness-related works like 3DFed [3] focus on various types of defense strategies and assume that the server stops training after the end of attacks. This paper primarily centers on enhancing backdoors’ stealthiness and effectiveness. However, to improve durability, we plan to explore combining the LP attack with other complementary attack strategies in our future work.
>
> [1] Zhang, Zhengming, et al. "Neurotoxin: Durable backdoors in federated learning." International Conference on Machine Learning. PMLR, 2022.
>
> [3] Li, Haoyang, et al. "3dfed: Adaptive and extensible framework for covert backdoor attack in federated learning." 2023 IEEE Symposium on Security and Privacy (SP). IEEE, 2023.
>
> Q4: A: For a fair comparison with existing works, we align with recent research in 2023 like 3DFed and Flip, which employ broader datasets such as CIFAR10 and Fashion-MNIST. We are running and will provide evaluations on the LEAF datasets in our revision later.

---

> > ### Comment · Reviewer_MrXZ · 2023-11-22
> >
> > Thank you for the detailed response.
> >
> > It is very difficult to understand why Neurotoxin will not evade any defense and the proposed method will without any results. The authors of Neurotoxin argue that their approach can efficiently inject durable backdoors while evading defenses by manipulating the number of poisoned parameters. Although the proposed method is novel compared to Neurotoxin, the ultimate goal remains the same.
> >
> > Moreover, most state-of-the-art attacks prioritize both stealth and durability. Therefore, concentrating solely on stealth at the expense of durability diminishes the significance of the paper.
> >
> > I would like to keep my rating.

---

> > > ### Author Response · Authors · 2023-11-23
> > >
> > > To explore the stealthiness of Neurotoxin, we further conduct experiments on Neurotoxin under stringent defenses. We consider FLAME and Multikrum as stringent defenses as they show the best performance in both Table 2 and Table A-6 in our paper. We attack ResNet18 with the IID CIFAR-10 dataset with the same settings in our paper using the hyperparameter $k=0.01$ and $k=0.1$, where $k=0.01$ is the default setting in Neurotoxin and Neurotoxin shows worse performance when $k>0.1$. The table below shows that Neurotoxin with $k=0.1$ and $k=0.01$ is not enough to evade stringent defenses, as it still updates a large number of parameters in the model. The distance between malicious models and benign models can still be large and easy to detect.
> > >
> > > To explore the durability of LP attack, we stop attacking after 200 rounds and keep training the global model ResNet18 in IID CIFAR-10 dataset under the FedAvg. The results show that BSR in both baseline attack and LP attack do not degrade after training 1000 rounds. Because the loss in local datasets is close to 0 after the convergence of the global model, the updates from benign models do not diminish the backdoor task. So we claim that our LP attack can improve stealthiness without sacrificing durability.
> > >
> > > 3DFed [1] also mainly focuses on the stealthiness of backdoor attacks, where the paper does not have much discussion on its durability.
> > >
> > > The LP attack is one of the applications of BC layers. The implications of BC layers and the potential of layer substitution analysis can be utilized for future exploration. One such prospect involves delving into the nuanced properties of backdoor tasks and understanding how neural networks respond to them. In our extensive examination, detailed in Section 16 of the Appendix, we observed that a prolonged training epoch leads to the dispersion of the backdoor task across a greater number of layers and that both convolutional and fully connected layers collaborate seamlessly in response to backdoor tasks within the context of CV models.
> > >
> > > [1] Li, Haoyang, et al. "3dfed: Adaptive and extensible framework for covert backdoor attack in federated learning." 2023 IEEE Symposium on Security and Privacy (SP). IEEE, 2023.
> > >
> > > | Model    | Attack     | k    | Defense   | BSR (%) | Acc (%) |
> > > | -------- | ---------- | ---- | --------- | ------- | ------- |
> > > | ResNet18 | LP attack  | -    | FedAvg    | 95.35   | 80.27   |
> > > | ResNet18 | Neurotoxin | 0.01 | FedAvg    | 94.9    | 80.56   |
> > > | ResNet18 | Neurotoxin | 0.1  | FedAvg    | 85.32   | 78.2    |
> > > | ResNet18 | LP attack  | -    | FLAME     | 89.41   | 75.85   |
> > > | ResNet18 | Neurotoxin | 0.01 | FLAME     | 3.86    | 76.72   |
> > > | ResNet18 | Neurotoxin | 0.1  | FLAME     | 3.68    | 77.86   |
> > > | ResNet18 | LP attack  | -    | Multikrum | 95.24   | 78.19   |
> > > | ResNet18 | Neurotoxin | 0.01 | Multikrum | 4.6     | 77.72   |
> > > | ResNet18 | Neurotoxin | 0.1  | Multikrum | 3.29   | 78.03   |
> > >
> > > As the discussion deadline is approaching, we just want to make sure that our response really makes sense to you. We are more than happy to discuss if there is still something still unclear. Can you update your score if appropriate? Thank you!

---

> > > > ### Comment · Reviewer_MrXZ · 2023-11-23
> > > >
> > > > I really appreciate the authors' clarification with further experiments. It mostly addressed my concerns. I have updated the score accordingly. I would request the authors to revise the manuscript with the new results and discussions.

---

> > > > > ### Author Response · Authors · 2023-11-23
> > > > >
> > > > > We sincerely appreciate that you could raise the score. We will update the manuscript with the new results and detailed analysis. Thanks.

---

### Official Review · Reviewer_vQhS · 2023-11-01

**Soundness:** 3 good
**Presentation:** 3 good
**Contribution:** 3 good
**Rating:** 6
**Confidence:** 4

**Summary:**

The paper proposes the concept of "backdoor-critical" (BC) layers, which are a small subset of model layers that dominate the model's vulnerability to backdoor attacks.
They introduce a method called Layer Substitution Analysis to identify BC layers from the attacker's perspective. This involves substituting layers between a benign model and malicious model and evaluating the impact on backdoor success rate.
Based on the identified BC layers, they design two new backdoor attack methods - layer-wise poisoning attack and layer-wise flipping attack. These precisely poison only the BC layers to inject backdoors while minimizing detectability. Experiments on CIFAR-10 and Fashion-MNIST datasets show their attacks can successfully bypass state-of-the-art defenses like Multi-Krum, FLAME, and RLR. The attacks achieve higher backdoor success rates and main task accuracy compared to prior attacks

**Strengths:**

1.	Designs two highly targeted poisoning attacks (layer-wise poisoning and flipping) that precisely exploit BC layers to inject backdoors. Requires minimal model modification.
2.	Comprehensive experiments show the BC layer-aware attacks can bypass state-of-the-art defenses like Multi-Krum, FLAME, RLR etc. Achieves higher attack success rate and main task accuracy.
3.	Analysis of BC layers provides new perspectives for future research into vulnerabilities of federated learning models and development of more robust defenses.
4.	Well-written paper with clear motivation, technical approach, extensive experiments and analysis. Meaningful insights for both attacks and defenses in federated learning.

**Weaknesses:**

1.	More analysis needed on why certain layers tend to be BC layers, and how factors like model architecture, data, triggers etc. influence this.
2.	Layer substitution analysis to identify BC layers has high computational overhead since it requires retraining models multiple times.
3.	How diversity across clients' data affects BC layer analysis needs more investigation. Paper assumes attacker has representative clean and poisoned data.
4.	No ablation study on key components of the layer-wise attacks like malicious model averaging, adaptive layer control etc.

**Questions:**

How do different trigger types (pixel, semantic, hardware-based) influence the BC layers? Can triggers be designed to target specific layers?

---

> ### Author Response · Authors · 2023-11-16
>
> Responses to Weaknesses:
>
> W1: A: Please refer to Section 16 of the Appendix, where we explored how various factors, such as the shape of triggers, types of backdoor attacks (types of triggers), model initial weights, training hyperparameters, and model architecture, influence the behavior of BC layers.
>
> W2: A: Please refer to Section 8 of the Appendix, where we provided the complexity analysis of our algorithm. LP attack provides two approaches to saving computing resources. Fig. A-16 in the Appendix shows that LP attack is capable of injecting a backdoor and achieving 60% BSR with as few as 0.02% of clients as malicious clients, indicating that the attacker can reduce computational costs by minimizing attack frequency. As Fig. 6 shows, LP attack remains effective and achieves 80% BSR when the attacker reuses BC layers identified in previous rounds and only identifies BC layers in each 10 rounds. Those capabilities allow attackers to balance the computational costs and the attack performance, based on their computing capabilities.
>
> W3: A: In our settings, malicious clients are randomly chosen from the entire client pool. Please refer to Section 15 of the Appendix, where we conducted experiments on various Non-iid levels. The results show that our LP attack remains effective and achieves 80% BSR even when confronted with datasets exhibiting lower diversity q=0.8.
>
> W4: A: Thanks for the suggestions. We will update the table in the ablation study in the revision as follows:
>
>   | Distribution | Model    | Average Model | Adaptive Control | MAR (%) | BSR (%) |
>   | ------------ | -------- | :-------------: | :----------------: | ------- | ------- |
>   | Non-IID      | ResNet18 | √             | √                | 93.0    | 90.7    |
>   | Non-IID      | ResNet18 | √             | ×                | 76.0    | 87.4    |
>   | Non-IID      | ResNet18 | ×             | √                | 66.5    | 93.4    |
>   | Non-IID      | ResNet18 | ×             | ×                | 51.8    | 87.6    |
>
>
> Responses to Questions:
>
> Q1: A: Please refer to Section 16 of the Appendix, where we explored how various factors, such as the shape of triggers, types of backdoor attacks (types of triggers), model initial weights, training hyperparameters, and model architecture, influence the behavior of BC layers.  With PGD and backward propagation, we can design triggers for targeting specific layers. We will leave it in our future work.

---

> ### Author Response · Authors · 2023-11-23
>
> As the discussion deadline is approaching, we just want to make sure that our response really makes sense to you. We are more than happy to discuss if there is still something still unclear. Can you take a look at the rebuttal and update your score if appropriate? Thank you!

---

### Official Review · Reviewer_tdce · 2023-11-04

**Soundness:** 2 fair
**Presentation:** 3 good
**Contribution:** 2 fair
**Rating:** 6
**Confidence:** 3

**Summary:**

The paper highlights a previously underexplored aspect of FL security: backdoor-critical (BC) layers within neural networks. Unlike conventional attacks that target the entire model, focusing on BC layers can lead to equally damaging outcomes with a significantly lower probability of detection by SOTA defense mechanisms.

The contribution is an in-situ approach Layer Substitution Analysis that enables attackers to identify and verify these BC layers. With this knowledge, the authors design 2 attacks: layer-wise poisoning attack and layer-wise flipping attack.

The proposed methodology is tested against SOTA FL defense strategies, demonstrating that even with as few as 10% of the participants in the FL system being malicious, the BC layer-aware attacks can successfully implant backdoors into the model.

The experiments show that these BC layer-aware attacks not only succeed in evading current defenses but also surpass the performance of the latest backdoor attack methods.

**Strengths:**

- The paper proposes a novel and interesting analysis to precisely identify backdoor-critical layers.
- The paper provides a comprehensive evaluation to show the effectiveness and stealthiness.

**Weaknesses:**

- The design only consider backdoor success rate, without considering the clean accuracy of the layer substitution, which is not reasonable.
- No discussion on the limitations.

**Questions:**

1. The proposed method only consider backdoor success rate, why not also consider the clean accuracy of the layer substitution? What if the layer substitution leads to a significant drop in clean accuracy?

2. Could the author explain the intuition behind the layer substitution? Compared with existing attacks, even with less backdoor-related layer, it keeps the backdoor behavior (even enhances it). Why can it be stealthier, surpassing the defense methods?

3. For sensitivity analysis for threshold $\tau$ in Section 4.5, the BSR increase with the increase of $\tau$, which is reasonable. But I am interested in how the MAR changes with the increase of $\tau$. This can illustrate how the number of substituted layer affects the stealthiness. Is there any trend?

4. For Step 2 in Section 3.2, I am curious what is the typical value range of $\Delta BSR_{b2m(l)}$? Is any statistics?

---

> ### Author Response · Authors · 2023-11-16
>
> Responses to Weaknesses:
> - No discussion on the limitations
>
> A: Thanks for your suggestions. We will add the following discussion as the Limitation Section in our revision:
> Single-shot attack (Bagdasaryan et al., 2020) has the capability to inject a backdoor into the global model through a malicious client within a single round by scaling the parameters of malicious models. While our LP attack can narrow the distance gap by targeting BC layers, we acknowledge that it may not effectively support a large scaling parameter, such as $\lambda = 100$ in DBA, when confronted with stringent distance-based defenses. However, there are several possible methods to improve the LP attack to support larger scaling parameters, e.g., searching BC neurons or designing triggers related to fewer parameters.
>
> Responses to Questions:
>
> Q1: A: As malicious models are trained on the dataset combined with clean data and poisoned data like normal backdoor attacks, the malicious models maintain high clean accuracy before layer substitution. If the server develops defenses taking clean accuracy as measurements to select models, our adaptive layer control mechanism can adjust the number of attacking layers. The reduction of the number of attacking layers can raise the clean accuracy of uploaded models and bypass the detection. We conduct an experiment in VGG19 trained on the CIFAR-10 dataset to demonstrate that the attacker can meet the requirement of clean accuracy by adjusting the number of attacking layers. The results in the table below show that more attacking layers cause higher BSR but lower clean accuracy. Thus, adaptive layer control can decrease the number of attack layers to bypass detection. Notably, this experiment is used to stimulate the decision choice for one malicious client within a single round. The BSR and clean accuracy in this experiment do not mean the BSR and clean accuracy in the global model in FL, as the attacker will perform LP attack for multiple rounds to increase BSR in the global model and the server will aggregate models from all clients to improve the clean accuracy in the global model.
>
> | Number of Attacking Layer | BSR (%) | Acc(%) |
> | --------------- | ------- | ------ |
> | 0               | 0.63    | 86.62  |
> | 1               | 3.01    | 86.36  |
> | 2               | 2.36    | 84.97  |
> | 3               | 2.7     | 84.0   |
> | 4               | 17.0    | 82.52  |
> | 5               | 28.67   | 78.36  |
> | 6               | 72.12   | 73.41  |
> | 7               | 90.7    | 69.3   |
> | 8               | 96.79   | 62.9   |
>
>
> Q2: A: Neural networks are inherently overparameterized and can be pruned down to subnetworks that meet or surpass the original network's test accuracy, e.g., the lottery ticket theory [1] [2].
>
> Intuitively, the BC layers can be seen as subnetworks so LP attack can keep and even potentially increase the BSR (similar to the lottery ticket). According to [3], compact hypotheses can better generalize.  Layer substitution, a pruning-like operation, can improve the generalization of backdoor tasks in the global model. The reason why LP attack can bypass defense methods is that the malicious models in LP attack carry fewer malicious parameters. This reduction in malicious parameters decreases the distance from benign models, enabling the LP attack to circumvent detection.
>
> [1] Frankle, Jonathan, and Michael Carbin. "The Lottery Ticket Hypothesis: Finding Sparse, Trainable Neural Networks." International Conference on Learning Representations. 2018.
>
> [2] Cheng, Yu, et al. "An exploration of parameter redundancy in deep networks with circulant projections." Proceedings of the IEEE international conference on computer vision. 2015.
>
> [3] Rissanen, Jorma. "Stochastic complexity and modeling." The annals of statistics (1986): 1080-1100.
>
> Q3: A: Results across all values of $\tau$ demonstrate that the MARs in LP attack consistently remain at high levels and surpass 90%. This achievement can be attributed to two mechanisms:  model averaging and adaptive control, which are introduced to ensure stealthiness in the LP attack. Model averaging keeps the uploaded malicious model close to the center of benign models. Adaptive layer control can decrease the number of attack layers if the malicious models deviate from benign models greatly. The ablation study presented in Table 4 underscores the significance of both mechanisms, revealing that the MARs may decrease from 93% to 52% in their absence.
>
> Q4: A: In ResNet18, three convolutional layers cause drops over 5% and linear.weight layer causes the highest drop (20%), while most layers have no impact on the BSR. In VGG19, six convolutional layers lead to decreases of more than 5% and feature.10.weight causes the highest drop (9.5%), while half of the weight layers cause no influence on BSR. We can add those figures to our revision if needed.

---

> ### Author Response · Authors · 2023-11-23
>
> As the discussion deadline is approaching, we just want to make sure that our response really makes sense to you. We are more than happy to discuss if there is still something still unclear. Can you take a look at the rebuttal and update your score if appropriate? Thank you!

---

> > ### Comment · Reviewer_tdce · 2023-11-23
> >
> > I appreciate the authors for their efforts in the rebuttal. My concerns are addressed and I raise the score accordingly.

---

> > > ### Author Response · Authors · 2023-11-23
> > >
> > > We truly appreciate that you raised the score, and we are very happy that our responses have addressed the issues. Thank you very much again for your constructive feedback on our paper.

---

### Meta-Review · Area_Chair_h2XY · 2023-12-12

**Metareview:**

The rebuttal addressed some concerns so after rebuttal, all reviewers rated this submission at level 6. hence, it will be accepted.

**Justification For Why Not Higher Score:**

No reviewer rated it above level 6, so I believe poster is the right decision. The main weaknesses are lack of sufficient discussion and limitations, lack of sufficient ablation study, and sufficient comparison with SOTA.

**Justification For Why Not Lower Score:**

I agree with the reviewers that this paper has solid contribution and can be accepted in ICLR.

---

### Decision · Program_Chairs · 2024-01-16

Accept (poster)